# Sarcoma classification by DNA methylation profiling

Sarcomas are malignant soft tissue and bone tumours affecting adults, adolescents and children. They represent a morphologically heterogeneous class of tumours and some entities lack defining histopathological features. Therefore, the diagnosis of sarcomas is burdened with a high inter-observer variability and misclassification rate. Here, we demonstrate classification of soft tissue and bone tumours using a machine learning classifier algorithm based on array-generated DNA methylation data. This sarcoma classifier is trained using a dataset of 1077 methylation profiles from comprehensively pre-characterized cases comprising 62 tumour methylation classes constituting a broad range of soft tissue and bone sarcoma subtypes across the entire age spectrum. The performance is validated in a cohort of 428 sarcomatous tumours, of which 322 cases were classified by the sarcoma classifier. Our results demonstrate the potential of the DNA methylation-based sarcoma classification for research and future diagnostic applications.

Sarcomas are a heterogeneous group of tumours, which pose challenges to pathologists. Many entities lack unequivocal morphologic or molecular hallmarks and the overall rarity of sarcomas result in a widespread lack of experience[1,2]. A high inter-observer variability among pathologists is reflected in considerable discrepancy rates between primary institutions and specialized referral centres with access to comprehensive molecular testing[3,4]. Pathologists often rely on the determination of tumour specific molecular alterations if available[4]. While the determination of characteristic molecular alterations most often consisting of translocations that generate gene fusions has become a diagnostic standard for many sarcoma types, approximately half of the sarcoma entities lack unequivocal molecular hallmarks[1]. Even in some cases defined by specific gene fusions, it may not be possible to identify adequately the fusion by FISH or RNA-based methods for a variety of technical and specimen-related limitations. Novel approaches are needed to fill these diagnostic gaps[5].

DNA methylation is a key epigenetic mark and plays important roles in normal development and disease[6]. In cancer, DNA methylation patterns reflect both the cell type of origin, as well as acquired changes during tumour formation[7]. Profiling of human brain tumours has demonstrated entity-specific methylation signatures and has led to the identification of several novel and clinically relevant subtypes[8–13]. On this basis, a comprehensive brain tumour classifier has been developed[14,15]. Recently, we have extended the principle of methylation-based tumour profiling to small blue round cell sarcomas evading a definite histological diagnosis, thereby resolving these cases into established sarcoma entities[16]. Further, DNA methylation-based profiling showed diagnostic potential for soft tissue and bone sarcoma subtyping[17–22]. In this work, we aimed at developing a DNA methylation-based classification tool for soft tissue and bone sarcomas representing a broad range of subtypes and age groups.

## Results

**DNA methylation profiling of prototypical sarcomas**. We subjected prototypical cases of the most common soft tissue and bone tumours, non-mesenchymal tumours that might mimic mesenchymal differentiation, i.e. squamous cell carcinoma or melanoma, and non-neoplastic control tissue to DNA methylation profiling using the Infinium HumanMethylation450K BeadChip or EPIC array platform. Following quality control, methylation data were analysed by unsupervised hierarchical clustering and t-Distributed Stochastic Neighbour Embedding (t-SNE)[23] thereby identifying groups of tumours sharing methylation patterns (methylation classes). To minimize potential clustering artefacts at least seven cases were required for defining a methylation class, which empirically proved sufficient for training a classifier and allowed prediction[14,15]. Unsupervised clustering, respecting the minimal number of seven cases per group, led to the designation of 62 tumour methylation classes belonging altogether to 54 histological types, and three non-neoplastic control methylation classes (Fig. 1). Iterative random down-sampling validated the stability of these methylation classes (Supplementary Fig. 1), and potential confounding factors such as sex, patients' age, type of material, type of array and tumour purity were excluded (Supplementary Fig. 2).

Based on 1077 tumour cases, methylation classes were assigned to four categories relating to the WHO classification (Fig. 1a). Category 1 represents methylation classes equaling a WHO entity. Category 2 represents methylation classes corresponding to a subgroup of a WHO entity. Category 3 represents methylation classes that combine WHO entities. Category 4 represents methylation classes of novel entities which are not yet

defined by the WHO classification (Fig. 1a). 48 methylation classes corresponded to distinct WHO entities (category 1) comprising 45 mesenchymal tumour entities, cutaneous melanoma, cutaneous squamous cell carcinoma and Langerhans cell histiocytosis. Nine methylation classes corresponded to subsets within WHO entities (category 2) with conventional chondrosarcoma dividing into four methylation classes, rhabdomyosarcoma with *MYOD1* alteration, plexiform neurofibroma, dedifferentiated chordoma and small blue round cell tumours with either *BCOR* alteration or *CIC* alteration. Three methylation classes combined WHO entities (category 3). The methylation class angioleiomyoma/myopericytoma and the methylation class atypical fibroxanthoma/pleomorphic dermal sarcoma each combined two entities, while the methylation class undifferentiated sarcoma contained undifferentiated (pleomorphic) sarcoma, myxofibrosarcoma and a fraction of pleomorphic liposarcoma, thereby providing further evidence that these sarcoma subtypes probably fall into a morphologic continuum of a single entity as suggested by previous genetic-based studies[24–26]. Two methylation classes point towards novel entities not yet defined by the WHO (category 4)[13,19]. The methylation class SARC (RMS-like) was identified in sarcomatous CNS tumours with various morphologic patterns not matching established tumour categories. Unifying features of cases mapping to this class are rhabdomyoblast-like cells and *DICER1* mutations[13]. Methylation class SARC (MPNST-like) was reported as a subset of malignant peripheral nerve sheath tumours[19]. Cases assigning to SARC (MPNST-like) present similar to MPNST, however, retain trimethylation at histone 3 lysine 27 (H3K27me3). In addition, based on 28 non-neoplastic tissue specimens methylation classes were established for non-neoplastic skeletal muscle, reactive soft tissue and leukocytes. Supplementary Data 1 provides basic clinical information for each individual case of these methylation classes. Supplementary Data 2 indicates characteristic clinical and molecular features for each methylation class.

**Development of the sarcoma classifier**. We next developed a classification tool, sarcoma classifier, using a Random Forest machine learning classification algorithm as described[14,27]. Cross-validation, an internal performance metric[15], of the sarcoma classifier provided an estimated error rate of 1.95% for raw scores and a discriminating power of 99.9% by area under receiver operating characteristic curve analysis. The low rate of misclassifications demonstrates the discriminating power of the classifier algorithm (Fig. 2, Supplementary Data 2). The discrepancies encountered at cross-validation predominantly occurred between the four methylation classes of conventional chondrosarcoma and between three methylation classes of sarcomas associated with *BCOR* alterations. Similar to the brain tumour classifier we introduced a methylation class family score combining these closely related methylation classes by adding up their respective prediction scores. This modification reduced the error rate at cross-validation to 0.65% for the raw scores. We employed a calibration algorithm transforming raw into calibrated scores thereby ensuring inter-class-comparability. This further allowed definition of a general cut-off score of 0.9 as a threshold for prediction to a specific methylation class (Supplementary Fig. 3)[14].

**Classifier performance validated in a clinical cohort**. Next, the sarcoma classifier performance was validated on 428 additional cases, mostly representing relapsed and refractory soft tissue and bone tumours, enrolled in the MNP2.0, PTT2.0, INFORM or NCT MASTER trials, which are focused on molecular analysis (Supplementary Data 3)[28–30]. The predicted methylation class by

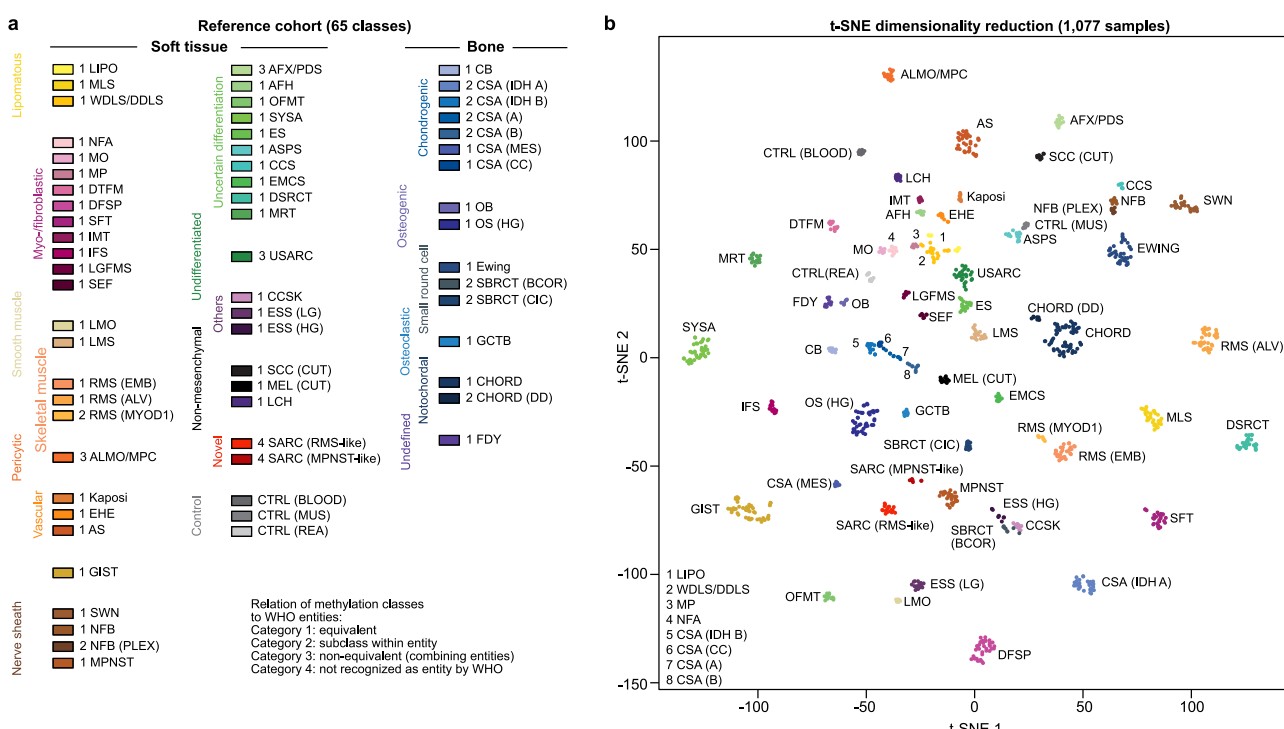

**Fig. 1 Establishing the DNA methylation-based sarcoma reference cohort. a** Overview of 62 tumour and three control DNA methylation classes included in the sarcoma classifier reference cohort. The methylation classes are colour-coded and grouped according to the WHO scheme. The relation between methylation classes and the WHO defined subtypes is categorised in 4 tiers: equivalent to a WHO entity (category 1); subgroup of a WHO entity (category 2); combining WHO entities (category 3); non-defined by WHO (category 4). **b** Visualisation of the reference cohort methylation profiles (n = 1,077) using t-distributed stochastic neighbour embedding (t-SNE) dimensionality reduction. Individual samples are colour-coded in the respective class colour (n = 65) as given in (**a**). Abbreviations: LIPO, lipoma; MLS, myxoid liposarcoma; WDLS/DDLS, well differentiated liposarcoma/dedifferentiated liposarcoma; NFA, nodular fasciitis; MO, myositis ossificans; MP, myositis proliferans; DTFM, desmoid-type fibromatosis; DFSP, dermatofibrosarcoma protuberans; SFT, solitary fibrous tumour; IMT, inflammatory myofibroblastic tumour; IFS, infantile fibrosarcoma; LGFMS, low-grade fibromyxoid sarcoma; SEF, sclerosing epithelioid fibrosarcoma; LMO, leiomyoma; LMS, leiomyosarcoma; RMS (EMB), embryonal rhabdomyosarcoma; RMS (ALV), alveolar rhabdomyosarcoma; RMS (MYOD1); rhabdomyosarcoma with *MYOD1* mutation; ALMO/MPC, angioleiomyoma/myopericytoma; EHE, epithelioid haemangioendothelioma; AS, angiosarcoma; GIST, gastrointestinal stromal tumour; SWN, schwannoma; NFB, neurofibroma; NFB (PLEX), plexiform neurofibroma; MPNST, malignant peripheral nerve sheath tumour; AFX/PDS, atypical fibroxanthoma/pleomorphic dermal sarcoma; AFH, angiomatoid fibrous histiocytoma; OFMT, ossifying fibromyxoid tumour; SYSA, synovial sarcoma; ES, epithelioid sarcoma; ASPS, alveolar soft part sarcoma; CCS, clear cell sarcoma of soft parts; EMCS, extraskeletal myxoid chondrosarcoma; DSRCT, desmoplastic small round cell tumour; MRT, malignant rhabdoid tumour; USARC, undifferentiated sarcoma; CCSK, clear cell sarcoma of the kidney; ESS (LG), low-grade endometrial stromal sarcoma; ESS (HG), high-grade endometrial stromal sarcoma; SCC (CUT), cutaneous squamous cell carcinoma; MEL (CUT), cutaneous melanoma; SARC, sarcoma; CTRL, control; MUS, muscle tissue; REA, reactive tissue; CB, chondroblastoma; CSA, chondrosarcoma; CSA (MES), mesenchymal chondrosarcoma; CSA (CC), clear cell chondrosarcoma; OB, osteoblastoma; OS (HG), high-grade conventional osteosarcoma; SBRCT, small blue round cell tumour; GCTB, giant cell tumour of bone; CHORD, chordoma; DD, dedifferentiated; FDY, fibrous dysplasia; LCH, Langerhans cell histiocytosis.

the sarcoma classifier was compared to institutional diagnoses (Fig. 3). A calibrated score ≥0.9 was reached for 322 of 428 cases (75%). The respective methylation class or -family matched with the institutional diagnosis in 263/428 cases (61%). A discrepant classifier prediction with a calibrated classifier prediction score ≥0.9 was encountered in 59/428 cases (14%). In these cases, molecular data were screened for subtype-specific alterations. The initial diagnosis was revised in favour of the predicted methylation class in 29/59 cases. In 26/59 cases the discrepancy between histological diagnosis and classifier prediction could not be resolved due to lack of entity specific mutations. The initial diagnosis was retained against the predicted methylation class in 4/59 cases (Fig. 4). The reason for misleading methylation class prediction in the latter cases, all passed the quality control steps, remains unclear. The 0.9 threshold was not reached for 106 of 428 cases (25%). Consecutive t-SNE analysis demonstrated a position of many of these cases peripheral or outside of the methylation classes from the reference set. It is possible that some of these tumours were contaminated with a higher amount of non-

neoplastic cells than estimated by histological examination, although the mean value for tumour cell purity of 47,4% in non-classifiable cases was only slightly lower compared to 51,3% in classifiable cases (Fig. 5). However, because some sarcomas with low calibrated classifier scores carried unique molecular alterations such as *ONECUT1-NUTM1* or *EWSR1-TFCP2* gene fusions we favour considering these as epigenetic subsets not yet covered by the current classifier version[31,32]. A heatmap for the performance of the classifier in the validation set is shown in Supplementary Fig. 4.

**Copy number profiling of sarcomas.** Independent from the methylation patterns used for classification, high-density DNA methylation arrays allow for determining copy number alterations, the detection of which is of major diagnostic relevance for sarcomas[25,26]. We generated copy number variation (CNV) plots from all sarcomas of the reference cohort as described[14]. Frequently encountered alterations include *MDM2* amplification for well-/dedifferentiated liposarcomas, *MYC* amplification for

radiation induced angiosarcoma or segmental chromosomal deletions on chromosome 22q encompassing *SMARCB1* for rhabdoid tumours. While these alterations often are characteristic for distinct sarcoma entities, they usually are not pathognomonic because of their occasional occurrence also in other entities. However, in combination with methylation profiles, CNV plots frequently add to the diagnostic decision process. The frequency of chromosomal or subchromosomal numerical alterations within the methylation classes/entities can be depicted by summary CNV plots (Supplementary Fig. 5). A systematic overview of frequently observed copy number alterations is provided for each methylation class (Supplementary Data 2). Molecular and clinical characteristics of the predicted methylation class are provided in a molecular classifier report (Supplementary Fig. 6).

## Discussion

We established an open-access platform allowing categorization of sarcomas based on machine generated methylation data and algorithm driven analysis. Employing DNA methylation-based categorization offers highly attractive features. Analyses can be performed on DNA extracted from paraffin-embedded and

formalin-fixed tissues allowing integration in routine settings. This represents a clear advantage over RNA expression profiling dependent on fresh tumour tissue[33]. The detection of individual methylation patterns for sarcoma entities is of special interest for those entities lacking pathognomonic gene alterations such as entity specific gene fusions. In the spectre of sarcomas currently recognized by the classifier approximately one third of the entities do not exhibit such specific mutational events.

Heterogeneity on DNA methylation level has been described between different tumours, but also within individual tumours for Ewing sarcoma[34]. On the other hand, that study also reported a close to 100% accuracy of distinguishing Ewing sarcoma from other cell types. Nevertheless, the observation of heterogeneity on the methylation level within individual tumours contrasts with the high stability of a parameter required for tumour classification. We here describe a high stability of methylation profiles for sarcoma entities. In addition, our selection process for CpG sites included in the classification algorithm favours those with maximal distinction between tumour entities. A practical example for the high stability of methylation profiles established by this approach has been presented for ependymoma with demonstration of primary and recurrent tumours from same patients neighbouring in almost all instances upon unsupervised clustering[9].

While conceptually highly attractive, the current version of the sarcoma classifier could not assign approximately 25% of the cases in the validation cohort to a DNA methylation class. This can be explained: Foremost, in its current stage the sarcoma classifier has not been trained to cover the entire spectrum of sarcoma subtypes. This does account for a portion of the 106/428 unrecognized cases exhibiting a calibrated score <0.9 (Fig. 3). Limited sample numbers for some entities will not allow identifying methylation subclasses as done for the chondrosarcomas splitting in four sub-categories. Future increase of the number of cases in the reference set will very likely enable detection of more methylation subgroups. A similar tendency has been observed in pilocytic astrocytomas and medulloblastomas separating now into several methylation subgroups with the clinical impact still remaining unclear[7,12,35]. Moreover, the DNA methylation-based approach is dependent on fairly high tumour cell content in the samples. Our experience is best with 70% or more of all cells in a sample constituting tumour cells[36]. Many sarcomas, however, typically contain high proportions of non-neoplastic inflammatory cells (Fig. 5). This circumstance might have contributed to classifier output scores lower than the cut-off score of 0.9, consequently prompting the tumour evaluation as unclassifiable. The effect of tumour cell purity on the classifier performance is likely

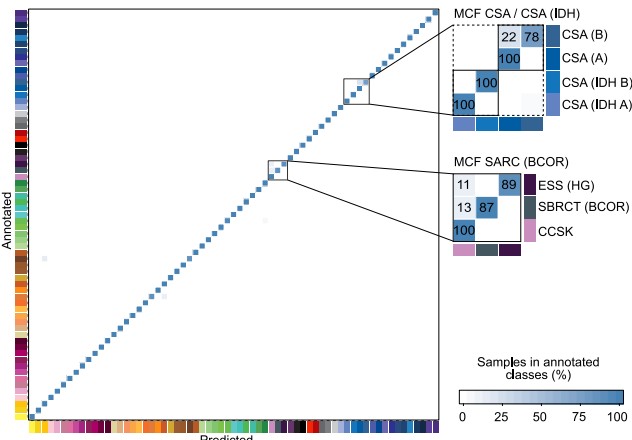

**Fig. 2 Cross-validation of the DNA methylation-based sarcoma classifier.** Heat map showing results of a threefold cross-validation of the Random Forest classifier incorporating information of n = 1077 biologically independent samples allocated to 65 methylation classes. Deviations from the bisecting line represent misclassification errors (using the maximum calibrated score for class prediction). Methylation class families (MCF) are indicated by black squares. The colour code and abbreviations are identical to Fig. 1a. Numbers of this figure are summarized in Supplementary Data 4.

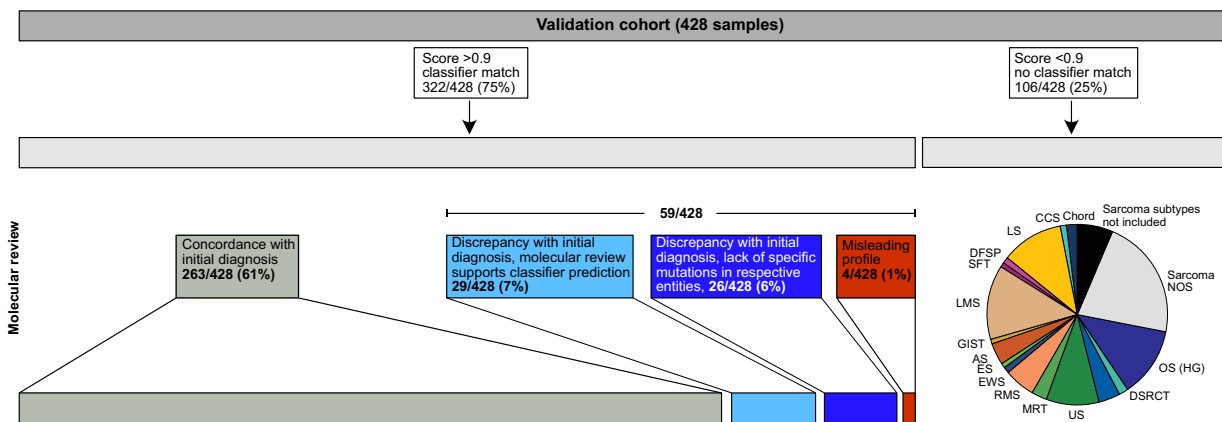

**Fig. 3 Validation of the sarcoma classifier.** In total, 426 independent sarcoma samples were analysed. 75% matched to an established DNA methylation class with a classifier prediction cut-off score of ≥0.9. 25% reached a classifier prediction cut-off score of <0.9. Abbreviations are identical to Fig. 1a.

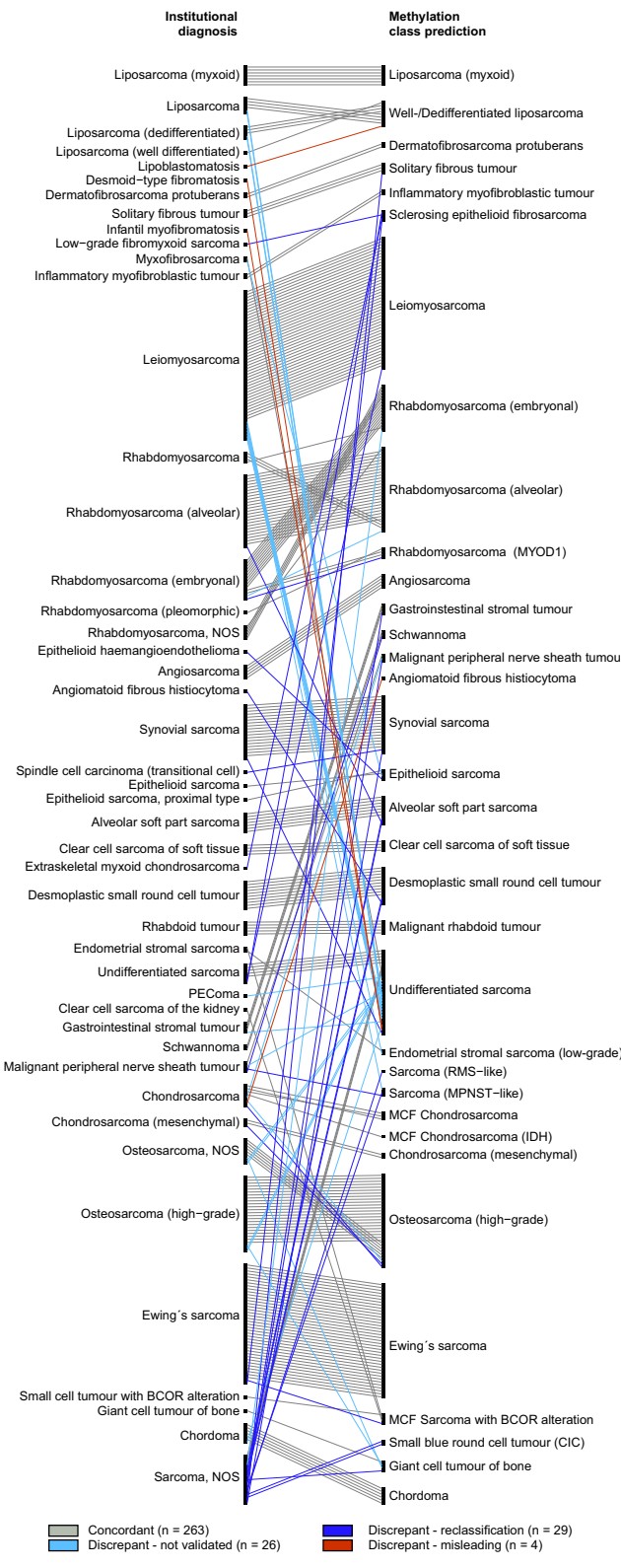

**Fig. 4 Comparison of pathological diagnosis and methylation class prediction.** Classifier validation using sarcoma cases enrolled in the MNP2.0, PTT2.0, INFORM or NCT MASTER trials. Institutional diagnosis (left) and classifier prediction (right) of the 322 cases that received a methylation class prediction ≥0.9. The institutional diagnosis of 263 cases matched the classifier prediction (concordant; grey bars). In 59 cases the classifier prediction differed from institutional diagnosis, with 29 cases reclassified in favour of the methylation class prediction (discrepant—reclassified; blue bars), 26 cases where molecular validation analysis was inconclusive (discrepant; light blue bars), and four cases with a misleading classifier result (discrepant – misleading; red bar).

performance, it likely would reduce the number of discordant cases as suggested by a recent study pointing to a reclassification rate of 14% in sarcoma upon central review[37].

In summary, we introduce a tool based on DNA methylation data and on automated algorithm analysis using probability measures for sarcoma classification. We developed a webpage for the scientific community listing characteristic features for the tumour methylation classes. This online platform also provides a free upload service for locally generated methylation data, which are analysed instantly and results are returned as molecular classifier report with a prediction confidence score (Supplementary Fig. 6). While the current version of the sarcoma classifier already includes some very rare entities, we acknowledge not to cover the entire spectrum. Analysis of additional sarcoma samples, including uploaded data, subject to permission, will further improve this tool by refining established and adding novel methylation classes. The sarcoma classifier can be accessed at www.molecularsarcomapathology.org.

## Methods

**Sample selection and quality control**. All samples of the reference and validation set are from individual/different patients. All cases of the reference set had undergone rigorous morphological examination by pathologists specialized in diagnosing sarcomas and also tumour-type specific molecular testing for identification of the relevant alterations, whenever possible. For each specimen, we aimed at a tumour cell content of ≥70%, with the caveat that microscopically estimated tumour cell percentage is prone to being relatively imprecise. However, determining tumour cell content by random forest regression demonstrated that this goal was not reached for many samples[38]. Our usual approach is the identification of a representative region on an H&E section followed by taking a 1.5 mm punch from the corresponding site in the formalin-fixed paraffin-embedded (FFPE) block. The validation set included sarcomas enrolled in the INFORM, NCT-MASTER, PPT and MNP2.0 studies[28–30]. Rare sarcoma entities have not been over-represented. However, availability determined inclusion resulting in over-representation of high-grade sarcomas in the validation set.

To exclude low-quality samples from the cohort, the on-chip quality metrics of all samples were checked and compared to a set of 7,500 pairs of IDAT-files. In addition, for each sample, an overall noise-level was computed using the R package conumee version 1.6.0. Samples showing low quality values ranging in the 10th percentile for at least one of the sample controls ('BC conversion I C1, C2, C3', 'BC conversion I C4, C5, C6' or 'BC conversion II 1, 2, 3, 4') and showing an overall noise level greater than 3, were excluded from this study.

**Methylation array processing**. All computational analyses were performed in R version 3.4.4 (R Development Core Team, 2019). Raw signal intensities were obtained from IDAT-files using the minfi Bioconductor package version 1.24.0. Illumina EPIC and 450k samples were merged to a combined dataset by selecting the intersection of probes present on both arrays (combineArrays function, minfi). Each sample was individually normalized by performing a background correction (shifting of the 5th percentile of negative control probe intensities to 0) and a dye-bias correction (scaling of the mean of normalization control probe intensities to 10,000) for both colour channels. Subsequently, a correction for the type of material tissue (FFPE/frozen) and array (450k/EPIC) was performed by fitting univariate, linear models to the log2-transformed intensity values (removeBatch-Effect function, limma package version 3.34.5). The methylated and unmethylated signals were corrected individually. Beta-values were calculated from the retransformed intensities using an offset of 100 (as recommended by Illumina).

Before further analysis was undertaken, the following filtering criteria were applied: removal of probes targeting the X and Y chromosomes (n = 11,551),

to be dependent on the sarcoma subtype (Fig. 5). Future studies with larger case numbers are required to elucidate the effect of tumour purity on classifier performance. A possibility to overcome this problem might be to subtract methylation patterns typical for lymphocytes thereby accentuating patterns of the respective sarcoma entities. And lastly, our validation cohort did not receive a centralized pathological reference review. While such centralized expert review would not affect the classifier

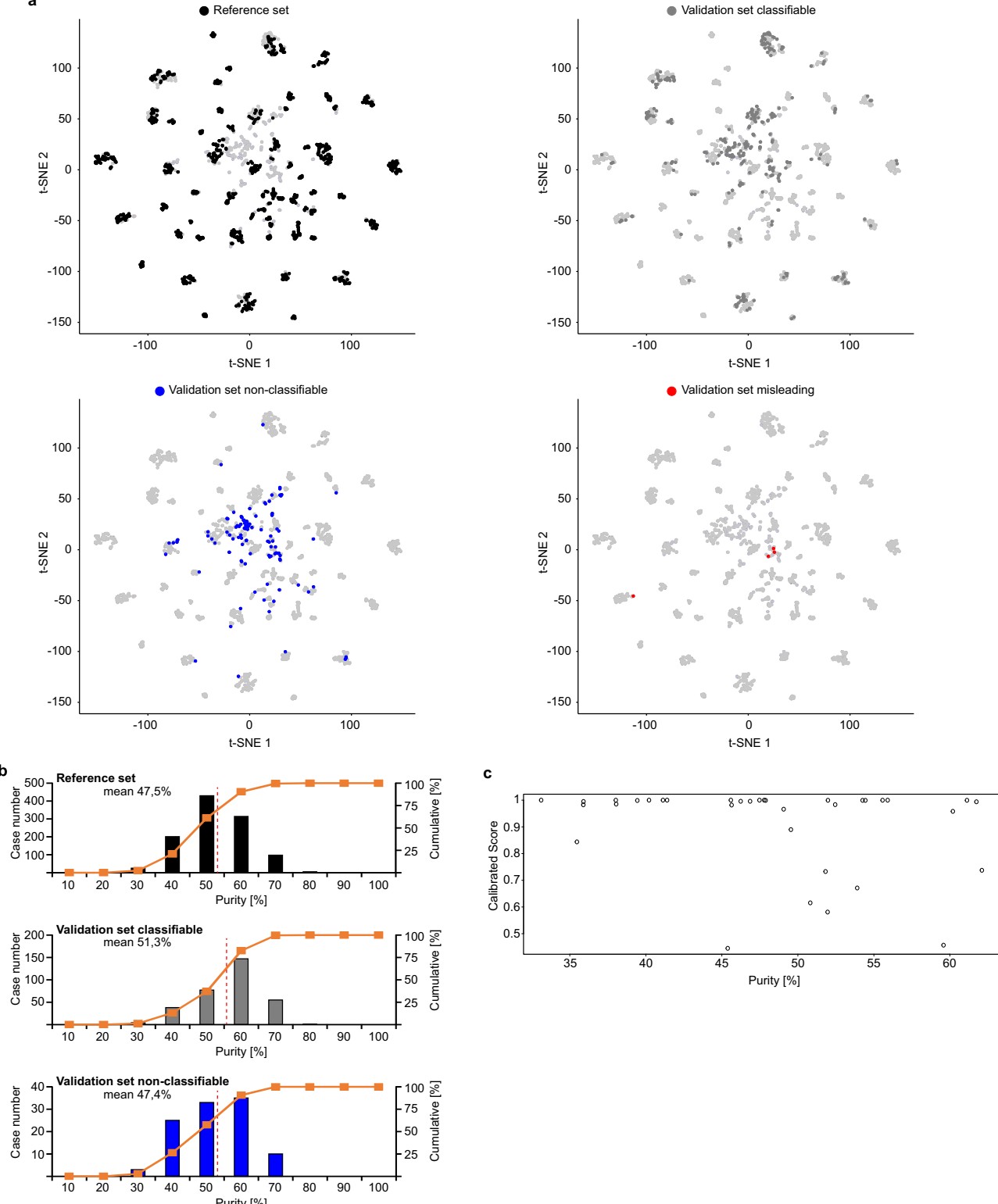

**Fig. 5 Impact of tumour cell purity on classifier performance. a** Unsupervised clustering of the combined reference ($n = 1077$) and diagnostic cohort ($n = 428$) using t-SNE dimensionality reduction. The reference set is indicated in the upper left plot. The diagnostic samples coded as classifiable ($n = 318$, grey dots; upper right plot), non-classifiable ($n = 106$, blue dots; lower left plot) and misleading ($n = 4$, red dots; lower right plot). The classifiable cases show high overlap with the reference cases. The non-classifiable cases frequently fall in the periphery of or are completely separate from the reference samples. **b** Tumour cell purity histogram plots of the reference set and the validation set subdivided into classifiable and non-classifiable cases. The mean value is indicated as dashed red line and provided as number [%]. **c** Tumour cell purity plotted against calibrated score for conventional osteosarcoma cases of the validation set.

removal of probes containing a single-nucleotide polymorphism (dbSNP132 Common) within five base pairs of and including the targeted CpG-site ($n = 7998$), probes not mapping uniquely to the human reference genome (hg19) allowing for one mismatch ($n = 3,965$), and 450k array probes not included on the EPIC array. In total, 428,230 probes were kept for downstream analysis.

### Unsupervised analysis

*t-SNE*. To perform unsupervised non-linear dimension reduction, the 10,000 most variable probes according to standard deviation were selected. The t-SNE plot was then computed via the R package Rtsne (version 0.13) using 3000 iterations and a perplexity value of 30. In addition, to assess the stability of the resulting projection, we repeated the t-SNE 500 times for subsamples of 90% of the data, sampled without replacement.

*Hierarchical clustering*. Unsupervised hierarchical clustering was performed using the 20,000 most variably methylated CpG sites across the dataset according to median absolute deviation, Euclidean distance and Ward's linkage method.

### Classifier development

Similar to the development of the brain tumour classifier[14] the Random Forest[27] algorithm (R package randomForest version 4.6-12) was applied to generate 10,000 binary decision trees, incorporating genome-wide information from all 1077 reference samples of the 65 methylation classes. We used the 10.000 CpGs with highest variable importance. In addition, to address unequal class size we performed downsampling as described[14]. The distribution of these CpGs position within the gene region and their regulatory feature group are indicated (Supplementary Fig. 7). Each binary decision tree assigns a given sample to one of the 65 classes, resulting in aggregate raw scores. To enable the comparison of classifier results between classes, these are transformed to a probability that measures the confidence in the class assignment (the calibrated score) by a L2-penalized multinomial logistic regression calibration model (R package glmnet version 2.0-18). Cross-validation of the Random Forest classifier resulted in an estimated error rate of 1.95% for raw scores and 0.65% for calibrated scores and a multi-class area under receiver operating characteristic curve[39] of 0.99 and a Brier score[40] of 0.05. This indicates a high discriminating power. To be able to classify samples from biologically closely related tumour classes, we introduced methylation class families. In those the calibrated scores were added to one score for the methylation class family[14].

### Classifier calibration

To obtain classifier scores that are comparable between classes and that are improved estimates of the certainty of individual predictions, we performed a classification score recalibration by mapping the original scores to more accurate class probabilities[15]. To find such a mapping, a L2-penalized, multinomial, logistic regression model was fitted, which takes the methylation class as the response variable and the Random Forest scores as explanatory variables. The R package glmnet[41] was used to fit this model. In addition, the model was fitted by incorporating a small ridge-penalty (L2) on the likelihood to prevent overfitting, as well as to stabilize estimation in situations in which classes are perfectly separable. Independent Random Forest scores are needed to fit this model, that is, the scores need to be generated by a Random Forest classifier that was not trained using the same samples, otherwise the Random Forest scores would be systematically biased and not comparable to scores of unseen cases. As such, Random Forest scores generated by the threefold cross-validation are used. To validate the class predictions generated by using the recalibrated scores of the calibration model, a nested threefold cross-validation loop is incorporated into the main threefold cross-validation that validates the Random Forest classifier[15]. Within each cross-validation run this nested threefold cross-validation is applied to generate independent Random Forest scores, which are then used to train a calibration model. The predicted Random Forest scores resulting from predicting the one-third test data of the outer cross-validation loop are then recalibrated by applying the calibration model that was fitted on the Random Forest scores generated during the nested cross-validation.

### Calibration model parameter tuning

To determine the optimal amount of L2-penalization for a calibration model a parameter tuning is performed using a resampling approach. To this end, each time a calibration model is fitted using raw RF scores from training data to calibrate raw RF scores from test data, 500 random data sets are generated by sampling 70% of the raw scores training data without replacement. For each of these random data sets, L2-multinomial logistic regression models were fitted applying a range of reasonable penalization parameters lambda. The remaining 30% scores were then calibrated by these models and maximum of the calibrated scores over all methylation classes was used to generate class predictions. Then a new binary class was defined, that is, predictions in agreement with the actual true class were considered 'classifiable' and predictions not in agreement were labelled 'non-classifiable'. This new binary variable and the accompanying maximum score over all class scores was then analysed by a receiver operator characteristics (ROC), i.e. calculating the Youden index (Specificity + Sensitivity − 1) for all possible thresholds. The final lambda was then determined such that the average Youden index over all resampling iterations at the prespecified cut-off threshold of 0.9 is maximal[15]. By tuning the calibration model in this way we can regulate the amount of calibration so that the scores perform well

at the prespecified common threshold of 0.9. This allows us to establish a common threshold for all forthcoming updates of the proposed classifier, which facilitates the communication with clinicians. A scheme summarizing the classifier algorithm steps is provided in Supplementary Fig. 8.

### Methylation class families

Misclassification errors mainly occurred within seven groups of histologically and biologically closely related tumour methylation classes. Therefore, we defined three 'methylation class families' (MCF) encompassing these seven tumour groups. Calibrated MCF score were calculated by summing up the calibrated class scores within one MCF.

### Estimating tumour purity from DNA methylation data

The estimated tumour purity for all reference cases was computed using the R package RF_Purify as described[38]. For the illustrations, the predictions obtained with the method 'ABSOLUTE' were used.

### Copy number profiling

Copy number alterations of genomic segments were inferred from the methylation array data based on the R-package conumee after additional baseline correction (https://github.com/dstichel/conumee). Summary copy number profiles were created by summarizing these data in the respective set of reference cases for each methylation class.

### Validation analysis

Cases enrolled in INFORM and NCT MASTER were subjected to total RNA and whole-exome sequencing; cases enrolled in MNP2.0 and PTT2.0 were subjected to a customized gene panel NGS[42] and total RNA sequencing from FFPE material[43], whenever necessary.

### Reporting summary

Further information on research design is available in the Nature Research Reporting Summary linked to this article.

## Data availability

Methylation data required for building the sarcoma classifier (reference set) were deposited at the public repository Gene Expression Omnibus under the accession number GSE140686. Supplementary Data 1 indicates the IDAT file names for each case. The remaining data are available within the Article, Supplementary Information or available from the authors upon request.

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

## Acknowledgements

This work was funded by Deutsche Krebshilfe grant 70112499, the NCT Heidelberg and an Illumina Medical Research Grant. Part of this work was funded by the National Institute of Health Research (to S.B. and Z.J.) and to UCLH Biomedical research centre (BRC399/NS/RB/101410). Human tissues were obtained from University College London NHS Foundation Trust as part of the UK Brain Archive Information Network (BRAIN UK, Ref: 18/004) which is funded by the Medical Research Council and Brain Tumour Research UK. The methylation profiling at NYU is supported by a grant from the Friedberg Charitable Foundation (to M.Sn.). M.Mi. would like to thank the Luxembourg National Research Fond (FNR) for the support (FNR PEARL P16/BM/11192868 grant).

## Author contributions

C.K. and A.v.D. conceived and supervised the project. C.K., D.Sc., D.St., M.Si., V.Ho., M.Sc., M.B.H., D.C., D.T.W.J., S.M.P. and A.v.D. performed DNA methylation data analysis and interpretation. C.K., D.Sc., D.St., M.Si., F.Sa., D.E.R., M.Bl., B.Wo., C.He., K.Be., P.Ho., S.Kr., E.Pf., S.St., P.Jo., F.Se., J.Ec., D.St., A.Re., A.K.W., P.Si., A.Eb., A.Su., F.F.K., B.Ca., A.Ko., F.K.F.K., M.Kr., M.Ko., A.St., B.B., H.G., C.Hei., W.H., D.T.W.J., S.F., S.M.P. and A.v.D. performed validation data analysis and interpretation. C.K., T.Mi., K.W.P., O.Wi., A.Ku., L.R.P., T.G.P.G., T.Ki., W.Wi., M.Pl., A.Un., M.Uh., A.Ab., J.De., B.Le., C.Th., M.Ha., W.Pa., C.Ha., O.St., M.Pr., J.He., S.Fr., Y.M.H.V., M.E.W., T.Me., K.Gr., E.d.A., J.D.M., M.A.I.G., K.T.C., S.Y.Y.L., A.Cu., M.Mi., M.My., S.Ru., U.Sc., V.F.M., J.Sc., J.Se., M.Sn., R.B., T.Kl., R.Bu., M.G., P.W., W.N.M.D., S.B., Z.J., Ly.I., P.S., O.M.T., M.Sa., J.M., J.A., X.G.M., S.M., M.E., J.K.B., M.L., E.W., C.A., A.F., U.D., P.H., D.B., C.V., G.M., U.F., I.P., S.F., S.M.P. and A.v.D. provided cases and meta data. C.K., D.Sc., D.St. and M.Si. created the figures. C.K., D.Sc., D.St., M.Si. and A.v.D. wrote the manuscript. The manuscript underwent an internal collaboration-wide review process.

## Funding

## Competing interests

A patent for a DNA methylation-based method for classifying tumour species of the brain has been applied for by the Deutsches Krebsforschungszentrum Stiftung des öffentlichen Rechts and Ruprecht-Karls-Universität Heidelberg (EP 3067432 A1) with S.M.P., A.v.D., D.T.W.J., D.C., V.Ho., M.Si., M.B.H. and M.Sc. as inventors. The other authors declare no competing interests.

## Additional information

Christian Koelsche [1,2,3,90], Daniel Schrimpf [1,2,90], Damian Stichel [2,90], Martin Sill [4,5,90], Felix Sahm [1,2], David E. Reuss [1,2], Mirjam Blattner [4,6], Barbara Worst [4,6,7], Christoph E. Heilig [8], Katja Beck [8,9], Peter Horak [8], Simon Kreutzfeldt [8], Elke Paff [4,6,7], Sebastian Stark [4,6,7], Pascal Johann [4,6,7], Florian Selt [4,7,10], Jonas Ecker [4,7,10], Dominik Sturm [4,6,7], Kristian W. Pajtler [4,5,7], Annekathrin Reinhardt [1,2], Annika K. Wefers [1,2], Philipp Sievers [1,2], Azadeh Ebrahimi [2], Abigail Suwala [1,2], Francisco Fernández-Klett [1,2], Belén Casalini [2], Andrey Korshunov [1,2], Volker Hovestadt [11,12], Felix K. F. Kommoss [3], Mark Kriegsmann [3], Matthias Schick [13], Melanie Bewerunge-Hudler [13], Till Milde [4,7,10], Olaf Witt [4,7,10], Andreas E. Kulozik [4,7], Marcel Kool [4,5], Laura Romero-Pérez [14], Thomas G. P. Grünewald [14], Thomas Kirchner [15], Wolfgang Wick [16,17], Michael Platten [18,19], Andreas Unterberg [20], Matthias Uhl [21,22], Amir Abdollahi [21,22,23,24], Jürgen Debus [21,22,23,24], Burkhard Lehner [25], Christian Thomas [26], Martin Hasselblatt [26], Werner Paulus [26], Christian Hartmann [27], Ori Staszewski [28,29], Marco Prinz [28,30,31], Jürgen Hench [32], Stephan Frank [32], Yvonne M. H. Versleijen-Jonkers [33], Marije E. Weidema [33], Thomas Mentzel [34], Klaus Griewank [35], Enrique de Álava [36,37], Juan Díaz Martín [36], Miguel A. Idoate Gastearena [38], Kenneth Tou-En Chang [39], Sharon Yin Yee Low [40], Adrian Cuevas-Bourdier [41], Michel Mittelbronn [41,42,43,44], Martin Mynarek [45], Stefan Rutkowski [45], Ulrich Schüller [45,46,47], Viktor F. Mautner [48], Jens Schittenhelm [49], Jonathan Serrano [50], Matija Snuderl [50], Reinhard Büttner [51], Thomas Klingebiel [52], Rolf Buslei [53], Manfred Gessler [54], Pieter Wesseling [55,56], Winand N. M. Dinjens [57], Sebastian Brandner [58,59], Zane Jaunmuktane [59,60], Iben Lyskjær [61], Peter Schirmacher [3], Albrecht Stenzinger [3], Benedikt Brors [62], Hanno Glimm [63,64,65,66], Christoph Heining [64,65,66], Oscar M. Tirado [67], Miguel Sáinz-Jaspeado [67], Jaume Mora [68], Javier Alonso [69], Xavier Garcia del Muro [70], Sebastian Moran [71], Manel Esteller [72,73,74,75], Jamal K. Benhamida [76], Marc Ladanyi [76], Eva Wardelmann [77], Cristina Antonescu [76], Adrienne Flanagan [78,79], Uta Dirksen [80,81], Peter Hohenberger [82], Daniel Baumhoer [83], Wolfgang Hartmann [84], Christian Vokuhl [85], Uta Flucke [86], Iver Petersen [87,88], Gunhild Mechtersheimer [3], David Capper [89], David T. W. Jones [4,6], Stefan Fröhling [8], Stefan M. Pfister [4,5,7] & Andreas von Deimling [1,2 ✉]

[1]Department of Neuropathology, Institute of Pathology, Heidelberg University Hospital, Heidelberg, Germany. [2]Clinical Cooperation Unit Neuropathology, German Cancer Consortium (DKTK), German Cancer Research Center (DKFZ), Heidelberg, Germany. [3]Department of General Pathology, Institute of Pathology, Heidelberg University Hospital, Heidelberg, Germany. [4]Hopp Children's Cancer Center Heidelberg (KiTZ), Heidelberg, Germany. [5]Division of Pediatric Neurooncology, German Cancer Consortium (DKTK), German Cancer Research Center (DKFZ), Heidelberg, Germany. [6]Pediatric Glioma Research Group, German Cancer Consortium (DKTK), German Cancer Research Center (DKFZ), Heidelberg, Germany. [7]Department of Pediatric Oncology, Hematology and Immunology, Heidelberg University Hospital, Heidelberg, Germany. [8]Division of Translational Medical Oncology, National Center for Tumor Diseases (NCT) Heidelberg and German Cancer Research Center (DKFZ), German Cancer Consortium (DKTK), Heidelberg, Germany. [9]Heidelberg Center for Personalized Oncology (HIPO), German Cancer Research Center (DKFZ), Heidelberg, Germany. [10]Clinical Cooperation Unit Pediatric Oncology, German Cancer Consortium (DKTK), German Cancer Research Center (DKFZ), Heidelberg, Germany. [11]Broad Institute of MIT and Harvard, Cambridge, MA, USA. [12]Department of Pathology and Center for Cancer Research, Massachusetts General Hospital and Harvard Medical School, Boston, MA, USA. [13]Genomics and Proteomics Core Facility, German Cancer Consortium (DKTK), German Cancer Research Center (DKFZ), Heidelberg, Germany. [14]Max-Eder Research Group for Pediatric Sarcoma Biology, Institute of Pathology, Faculty of Medicine, LMU Munich, Munich, Germany. [15]Institute of Pathology, Faculty of Medicine, LMU Munich, Munich, Germany. [16]Neurology Clinic and National Center for Tumor Diseases, University Hospital Heidelberg, Heidelberg, Germany. [17]Clinical Cooperation Unit Neurooncology, German Cancer Consortium (DKTK), German Cancer Research Center (DKFZ), Heidelberg, Germany. [18]Department of Neurology, Mannheim University Medical Center, University of Heidelberg, Mannheim, Germany. [19]Clinical Cooperation Unit Neuroimmunology and Brain Tumor Immunology, German Cancer Consortium (DKTK), German Cancer Research Center (DKFZ), Heidelberg, Germany. [20]Department of Neurosurgery, Heidelberg University Hospital, Heidelberg, Germany. [21]Department of Radiation Oncology, Heidelberg University Hospital, Heidelberg, Germany. [22]Heidelberg Institute of Radiation Oncology (HIRO), National Center for Radiation Research in Oncology (NCRO), Heidelberg, Germany. [23]Heidelberg Ion-Beam Therapy Center (HIT), Heidelberg, Germany. [24]Translational Radiation Oncology, German Cancer Consortium (DKTK), National Center for Tumor Diseases (NCT), German Cancer Research Center (DKFZ), Heidelberg, Germany. [25]Department of Orthopaedics, Trauma Surgery and Paraplegiology, Heidelberg University Hospital, Heidelberg, Germany. [26]Institute of Neuropathology, University Hospital Münster, Münster, Germany. [27]Department of Neuropathology, Institute of Pathology, Hannover Medical School (MHH), Hannover, Germany. [28]Institute of Neuropathology, Faculty of Medicine, University of Freiburg, Freiburg, Germany. [29]Berta-Ottenstein-Programme for Clinician Scientists, Faculty of Medicine, University of Freiburg, Freiburg, Germany. [30]Signalling Research Centers BIOSS and CIBSS, University of Freiburg, Freiburg, Germany. [31]Center for Basics in NeuroModulation (NeuroModulBasics), Faculty of Medicine, University of Freiburg, Freiburg, Germany. [32]Department of Neuropathology, Institute of Pathology, Basel University Hospital, Basel, Switzerland. [33]Department of Medical Oncology, Radboud University Medical Center, Nijmegen, The Netherlands. [34]Dermatopathology Bodensee, Friedrichshafen, Germany. [35]Department of Dermatology, University Hospital Essen, West German Cancer Center,

University Duisburg-Essen, Essen, Germany. [36]Department of Pathology, Institute of Biomedicine of Sevilla (IBiS), Virgen del Rocio University Hospital, CSIC/University of Sevilla/CIBERONC, Seville, Spain. [37]Department of Normal and Pathological Cytology and Histology, School of Medicine. University of Seville, Seville, Spain. [38]Department of Pathological Anatomy, Clínica Universidad de Navarra, University of Navarra, Pamplona, Spain. [39]Department of Pathology and Laboratory Medicine, KK Women's and Children's Hospital, Singapore, Republic of Singapore. [40]Department of Neurosurgery, National Neuroscience Institute, Singapore, Republic of Singapore. [41]National Center of Pathology (NCP), Laboratoire National de Santé (LNS), Dudelange, Luxembourg. [42]Luxembourg Center of Neuropathology (LCNP), Luxembourg, Luxembourg. [43]Luxembourg Centre for Systems Biomedicine (LCSB), University of Luxembourg, Luxembourg, Luxembourg. [44]Department of Oncology (DONC), Luxembourg Institute of Health (LIH), Luxembourg, Luxembourg. [45]Department of Pediatric Hematology and Oncology, University Medical Center Hamburg-Eppendorf, Hamburg, Germany. [46]Institute of Neuropathology, University Medical Center Hamburg-Eppendorf, Hamburg, Germany. [47]Research Institute Children's Cancer Center Hamburg, Hamburg, Germany. [48]Department of Neurology, University Medical Center Hamburg-Eppendorf, Hamburg, Germany. [49]Department of Neuropathology, Institute of Pathology and Neuropathology, University Hospital of Tübingen, Tübingen, Germany. [50]Department of Pathology, New York University School of Medicine, New York, NY, USA. [51]Institute of Pathology, Cologne University Hospital, Cologne, Germany. [52]Department of Pediatric Hematology and Oncology, University Children's Hospital, Frankfurt/Main, Germany. [53]Institute of Pathology, Sozialstiftung Bamberg, Klinikum am Bruderwald, Bamberg, Germany. [54]Theodor-Boveri-Institute/Biocenter, Developmental Biochemistry, Würzburg University, Würzburg, Germany. [55]Princess Máxima Center for Pediatric Oncology, Utrecht, The Netherlands. [56]Department of Pathology, Amsterdam University Medical Centers/VUmc, Amsterdam, The Netherlands. [57]Department of Pathology, Erasmus MC Cancer Institute, Rotterdam, The Netherlands. [58]Department of Neurodegeneration, Institute of Neurology, University College London, London, UK. [59]Division of Neuropathology, The National Hospital for Neurology and Neurosurgery, University College London Hospitals NHS Foundation Trust, London, UK. [60]Department of Molecular Neuroscience, UCL Queen Square Institute of Neurology, University College London, London, UK. [61]Research Department of Pathology, University College London, London, UK. [62]Division of Applied Bioinformatics, German Cancer Consortium (DKTK), German Cancer Research Center (DKFZ), Heidelberg, Germany. [63]Translational and Functional Cancer Genomics, National Center for Tumor Diseases (NCT) and German Cancer Research Center (DKFZ), Heidelberg, Germany. [64]Department of Translational Medical Oncology, National Center for Tumor Diseases (NCT) Dresden and German Cancer Research Center (DKFZ), Dresden, Germany. [65]Center for Personalized Oncology, National Center for Tumor Diseases (NCT) Dresden and University Hospital Carl Gustav Carus Dresden at TU Dresden, Dresden, Germany. [66]German Cancer Consortium (DKTK), Dresden, Germany. [67]Sarcoma Research Group, Oncobell Program, Bellvitge Biomedical Research Institute (IDIBELL), CIBERONC, Barcelona, Catalonia, Spain. [68]Department of Pediatric Onco-Hematology and Developmental Tumor Biology Laboratory, Hospital Sant Joan de Déu, Barcelona, Catalonia, Spain. [69]Pediatric Solid Tumor Laboratory, Human Genetic Department, Research Institute of Rare Diseases, Instituto de Salud Carlos III (ISCIII), Madrid, Spain. [70]Medical Oncology Service, Catalan Institute of Oncology (ICO), Bellvitge Biomedical Research Institute (IDIBELL), University of Barcelona, Barcelona, Catalonia, Spain. [71]Cancer Epigenetics and Biology Program (PEBC), Bellvitge Biomedical Research Institute (IDIBELL), Barcelona, Catalonia, Spain. [72]Josep Carreras Leukaemia Research Institute (IJC), Badalona, Barcelona, Catalonia, Spain. [73]Centro de Investigacion Biomedica en Red Cancer (CIBERONC), Madrid, Spain. [74]Institucio Catalana de Recerca i Estudis Avançats (ICREA), Barcelona, Catalonia, Spain. [75]Physiological Sciences Department, School of Medicine and Health Sciences, University of Barcelona (UB), Barcelona, Catalonia, Spain. [76]Department of Pathology, Memorial Sloan Kettering Cancer Center, New York, NY, USA. [77]Gerhard Domagk Institute of Pathology, University Hospital Münster, Münster, Germany. [78]Department of Histopathology, Royal National Orthopaedic Hospital NHS Trust, Stanmore, Middlesex, UK. [79]University College London Cancer Institute, London, UK. [80]Pediatrics III Pediatric Hematology, Oncology, Immunology, Cardiology, Pulmonology, West German Cancer Center, University Hospital Essen, Essen, Germany. [81]International Ewing Sarcoma Study Group, German Cancer Consortium (DKTK), West German Cancer Center (WTZ), University Duisburg-Essen, Essen, Germany. [82]Division of Surgical Oncology and Thoracic Surgery, Mannheim University Medical Center, University of Heidelberg, Mannheim, Germany. [83]Bone Tumour Reference Centre at the Institute of Pathology, University Hospital Basel and University of Basel, Basel, Switzerland. [84]Division of Translational Pathology, Gerhard Domagk Institute of Pathology, University Hospital Münster, Münster, Germany. [85]Department of Pediatric Pathology, University Hospital of Schleswig-Holstein, Kiel, Germany. [86]Department of Pathology, Radboud University Medical Center, Nijmegen, The Netherlands. [87]Institute of Pathology, SRH Poliklinik Gera GmbH, Gera, Germany. [88]Institute of Pathology, Jena University Hospital Jena, Germany. [89]Department of Neuropathology, Charité Universitätsmedizin Berlin, Berlin, Germany. [90]These authors contributed equally: Christian Koelsche, Daniel Schrimpf, Damian Stichel, Martin Sill. ✉email: andreas.vondeimling@med.uni-heidelberg.de

