## [Peer Review File · Nature Communications]

Reviewers' Comments:

Reviewer #1:

Remarks to the Author:

The study by Koelsche et al presents a DNA methylation-based classifier of soft tissues and bone tumors. The 'sarcoma classifier' was trained on methylation profiles from 1,077 tumor samples from a reference sarcoma cohort. Methylation profiling was done using the EPIC and 450k Illumina arrays. Unsupervised hierarchical clustering and t-SNE analysis revealed 62 tumor-methylation classes belonging to 54 histological types. Using this dataset and a Random Forest machine learning classification, the authors developed the sarcoma classifier similarly to what they have done previously for tumors of the central nervous system (Capper et al., 2018). The classifier was validated on a set of 428 additional tumor samples coming mostly from relapsed/refractory tumors. Finally the authors used the DNA methylation profiles for detecting copy number alterations in the corresponding tumor samples.

Given that the authors have published a number of papers with similar focus previously (Capper et al., Nature, 2018; Koelsche et al., Clin Sarcoma Res, 2019; Koelsche et al., Journal of Clin Res and Clin Onc, 2019; Koelsche et al., Modern Pathology, 2018) – with the current manuscript having an almost identical figure lay out as the Capper et al paper – the added value of the work presented herein should be further emphasised.

What is the highlight of the presented work?

- New, better classifier?
- Identification of novel sarcoma types/subtypes?
- New diagnostic tool that could complement/replace current tumor-type specific molecular testing?

The potential of the presented sarcoma classifier as a diagnostic tool may be the most interesting/relevant aspect. Also, given that currently a lot of diagnostic centers perform RNA-seq for all newly diagnosed sarcomas, it would be interesting to assess how these two approaches perform in comparison.

Specific comments:

1. Line 199: 'recognised' may not be so precise. Replace with 'classified'
2. Method section corresponding to the analysis for Fig 1 is incomplete. The description of hierarchical clustering is missing. What do the authors mean by 'iteratively analysed'? How was the cluster number chosen? Why a minimum of seven cases was used (why not more or less)? How was the perplexity chosen?
3. Lines 386 – 391: This paragraph is difficult to read – not correct English. Please rephrase/restructure.
4. Figure 1b – Legend with sarcoma type abbreviations is really hard to read. Perhaps present this information as a table/figure. Or at least tumor types can be listed in alphabetical order.
5. Line 382: We aimed at a tumor cell content of >70%. However, according to Extended Figure 2f the predicted tumor purity for most samples is much lower.
6. Line 241/242 (Extended data Figure 2) – it is not clear what it is meant that potential confounding factors were excluded? Which factors?
7. Figure 1. The authors could explain a bit better what they mean by the four WHO categories. Do they mean the types of sarcomas based on WHO classification?
8. The task of the classifier is to split samples in methylation classes that were defined by unsupervised learning on the same dataset. For every sample that is tested in the cross-validation step of the Random Forest, the algorithm predicts to which of the 65 classes it belongs. However, these 65 classes have been defined using methylation profiles of all samples, including the profile of the tested sample. It should therefore be not surprising that the performance is high (0.999), as in a way, what is tested is the capability of the RF classifier to reproduce what the unsupervised learning did using the same data. It should be discussed if and how this biases the performance metrics, and if

and why this is still a reasonable comparison. This does not apply to the validation set, which is independent. Also, the methylation classes are biologically meaningful, as seen by the large overlap to WHO classification, but this issue should at least be discussed regardless.

9. Extended Figure 1a: λ Youden is shown as a line (not red dot)

10. Validation cohort: Are all classes evenly included in the validation cohort (an overview statement would be informative). Also, reference set is mostly based on diagnostic samples, whereas validation cohort is mostly relapsed tumors. Is there any overlap between diagnostic and relapsed tumors (coming from the same patient)?

11. Line 296: Typo mistake – 26/59? What does ‘remain open’ mean?

12. Figure 3 is not easy to follow. The 428 samples are relapsed/refractory samples. Why the top bar says Diagnostic samples? Where the 59 samples discussed in line 294 fit best in the figure?

13. It would be interesting to get an idea of the classification performance also on the samples that do not reach the 90% certainty threshold. This information is perhaps provided in Extended figure 3 but it is not discussed.

14. How is the performance for cases in which the classifier is very sure? E.g. at 99% certainty? What is the distribution of certainties in cases that the classifier did and did not agree with the diagnosis?

15. Given that CNAs have diagnostic relevance (as indicated in line 311) why CAN data are not included in the classifier? It may be interesting to compare its performance with vs without CNAs.

16. Lines 348-349: ‘portion of the unrecognized cases’ - it is not clear which cases the authors mean.

17. Lines 339-340: The illumina methylation arrays do not only cover promoter regions. The authors comment that their study focuses on CpG sites localized on promoter regions. Have they tested which of the CpG sites are the most informative for their classifier? They could add an additional analysis to test this. They could group the CpGs based on those found on core promoters, extended promoters, DNase I sites, gene bodies, regulatory elements etc and run the classifier using only a set of the CpGs and compare it to the performance of the classifier when they use all CpGs. This analysis may contribute to biological understanding for different sarcoma types.

18. How does the classification performance change with tumor impurity? This is important to assess and seems relevant for the end user. e.g. split the validation data in a few bins, based on the impurity, and assess performance there. If this cannot be done, at least significance should be assessed in Fig 5b. Also, there seem to be correlations between tumor type and impurity (Extended Figure 2); does this affect the classifier?

19. Discussion: What is the advantage of methylation based analysis (in addition to or instead of pathological assessment) in the end? Why should the end user consider doing methylation assays on top? It could uncover misclassifications, potentially altering treatment? However, at the moment it seems unsure if we could trust methylation enough to re-classify the tumor without additional genetic evidence for the new classification. So why not just do genetic classification on top of pathology, and save the time/money of methylation? How about RNA-seq that is currently used in clinics (at least for fusion driven sarcomas)? What are next steps to establish this in the clinic?

20. Classifier score calibration: REF 15 says linear SVMs in combination with MR worked best. Why not use that here?

21. Lines 446-448: does this mean that only samples which reach a threshold of 0.9 are taken into account for calculating the Youden index? Where do the 500 samples come from? Training or test set?

22. Classifier score calibration (page 18): a scheme should be provided, and the text structured in a way that makes it easier to follow. The scheme should include the complete CV scheme, and a visualization of the resampling and definition of the binary classes "classifiable" and "unclassifiable".

23. Figure 4. Color code should match Figure 3. Also, lines are hard to follow.

24. Figure 5. Most of the unclassified cases are located in the center of the plot. Is there an explanation for this?

25. Extended Figure 1b: it needs better labeling - x and y coordinates / iterations are not shown.

Reviewer #2:

Remarks to the Author:

Summary

The authors present a classification model that classifies soft tissue and bone sarcomas from 65 methylation groups. They used tSNE to identify classes that share methylation patterns. A random forests model was created to classify the 65 methylation groups using methylation data. The raw score obtained from the random forest model was then calibrated using a logistic regression model with L2 norm to obtain scores that were comparable between classes. Some of the methylation groups did not classify well, but seemed to be related to other methylation groups that also did not classify well. Similar methylation groups that did not classify well were combined into methylation class families and the classifier was able to correctly classify cases belonging to these families to the family, if not to the specific methylation group. The classification model was developed using the same methodology as in the paper "DNA methylation-based classification of central nervous system tumours" (Capper et al. 2018).

The authors verified the model using cross-validation and were able to achieve an error rate of less than 1% once all of their adjustments were made. They also verified their model using a validation data set. The error rate for this validation set was far greater. Twenty-five percent of the validation samples were not able to be classified by the model using the classification criterion. Of those that were classified, 61% were classified correctly. The authors investigated the cases that were not assigned a class and those that were classified incorrectly. Those that were not classified correctly were examined for subtype alterations and the additional information allowed them to correctly classify for about half of the misclassified cases. The cases that were not given a classification by the model did not cluster tightly within a methylation class in the tSNE analysis.

The authors also examined copy number variations (CNVs) and provided a table with the CNVs alterations for each methylation group, but that table was not included in the copy that I reviewed so I cannot comment on how helpful the CNV is for the overall tumor classification process.

Main concerns

The details of the classifier should be tightened up. It seems that the authors used the same method as the paper, "DNA methylation-based classification of central nervous system tumours" (I refer to this as the CNS paper below), but some of the important details are not clarified in your paper. The paper would improve if the following are clarified or if it is clarified which methodology on the CNS paper was followed exactly:

Where the number of probes that were used for the classifier trimmed? The CNS paper used variable importance to trim the number of probes down to 10,000.

Was down-sampling used to address unequal class sizes in the reference data set? It is mentioned that down-sampling was used, but did not indicate that it was used to address unequal class size as it was in the CNS paper.

What measure was used for the random forest score? The CNS paper defines the score as the proportion of times that a sample is classified as a certain methylation group in the 10,000 trees. The score can also refer to the F1 score, which is an entirely different metric. It is important that the reader is clear on how the raw score was calculated for your model.

More detail is needed in the description of the random forest scores used as the explanatory variables for the calibration model (logistic regression with L2 norm). In the CNS paper, it seems that the scores for a particular case are the random forest score for each methylation class. If the same method was used, there are 65 scores for each case, most of which will be 0. (I recognize that it may be less than 65 when the methylation families are taken into account).

Provide some additional discussion on why 0.9 was selected as a threshold for classification.

The paper notes that the samples in the reference dataset comprised at least 70% tumor cells, but this was not the case in the validation dataset. A table or figure that includes the information about the purity of the samples in the validation dataset, the raw and calibrated scores, and whether or not

the tumor was misclassified or not classified would be helpful and more clearly illustrate the effect the tumor purity may be having on the classifier. Figure 5b shows a graph of the purity of each of the samples, but does not provide the additional information needed to clearly determine the effect of the tumor purity on classification.

The discussion on the validation dataset states, "molecular data were screened for subtype specific alterations" and that the authors were able to correctly classify about half of the previously misclassified points using that information. Can this process be incorporated into the classifier? That is, if a tumor is misclassified, it would not be known in real life. How would the person making the diagnosis know that they should look for the subtype alterations and use that information to adjust the classifier assigned by the model? Using the numbers provided in the "Classifier performance validated in a clinical cohort" section, the model was only able to correctly classify about 46% of the validation cases ($428 * 0.75 * 0.61 = 195$, $195/428 = 0.4575$). If other information can be used in conjunction with the classifier to get to a correct diagnosis, it should be more clearly stated.

The calibration model: there are many other possible ways to normalize your model besides an L2 norm. It is not clear that other norms, such as the L1 norm or a combination of L1 and L2, would yield a higher classification rate, but it may be worth investigating other normalization methods to see if the non-classification and mis-classification rates can be decreased.

Please expand on how the copy number profiling data contributes to diagnosis in the context of the analysis done in this paper. Did it shed some light on some of the cases that were not classified or were not classified correctly? Illustrate a case or pattern that shows the merit of this additional information. I was not able to see Extended Data Figure 4 or Supplementary Table 2 because they were not included in my review copy so such examples may be present in those resources. However, even if that information is illustrated in the table and/or figure, pointing it out with a sentence or two in your text would illustrate the value of using the CNV information in conjunction with your classifier to diagnose a tumor.

Figure 2: This figure shows that the classifier is classifying very well, but because it is performing well, the figure is not providing very helpful information. The misclassified areas are difficult to identify because the color is so light. The numbers would be more helpful than the heat map in this situation. You may want to summarize this information in a table. A table can show each methylation group and the misclassification rates within each group for each classification stage (raw scores and calibrated scores) and indicate how closely related the misclassified point is to the predicted methylation class. The heat map may be very informative for the validation data. This is where you have many non-classification and misclassification errors. This type of heatmap may shed some light on the nature of these errors in that context. A table would also be helpful for the validation data. Figure 3 does provide additional information in regard to what is misclassified and a table would provide more detailed information.

Figure 4: This graph contains a lot of good information, but is hard to digest. If possible, it would be nice if the labels on the right lined up with the labels on the left. This allows the reader to quickly observe overall patterns, in particular when cases are correctly classified or erroneously classified to a methylation class that is closely related to the true methylation class. It would also be helpful to group the labels together in the same way they are in Figure 1 rather than in alphabetical order. That is, have tumor types from the same WHO category grouped together and incorporate the same color structure used in Figures 1 and 2 to further illustrate methylation classes that are closely related to each other.

Figure 5: This figure is an important illustration of the data, but it is a little hard to interpret all of the relationships it is trying to show. The dark color scheme with the red and gray dots was hard to visually interpret. I also would like to see the points that were misclassified in the validation dataset on the tSNE plot. The graphic may be more clear if you have four plots that show the same tSNE clusters for the reference dataset that are grayed out, and only add one color to each plot to highlight a single feature. The first plot can show the points that were misclassified in the reference dataset with one color, the second plot can highlight the points that were misclassified in the validation dataset in another color, the third plot can highlight the points that failed to be classified in the validation set with a third color, and the fourth plot can be the points that were classified correctly in the validation set with a fourth color. If these plots are gridded together, the relationships should be

easier to identify.

Minore comments

Line 221: Add a comma after "On this basis"

Line 225: Add a comma after "With this study"

Line 296: I believe it should be 26/59 cases

Line 318: Consider a comma after profiles

Reviewer #3:

Remarks to the Author:

This paper provides a more detailed DNA methylation profiling to classify sarcomas. Unlike previous work published by this group (Koelsche C, Clinical Cancer Research 2019 and Koelsche C, Mod Pathology 2018) this classifier was created by a much larger group of diverse sarcoma tumor types. They also had a smaller but still very large set for validation. The field of DNA methylation classification is very new in sarcomas compared to the more developed data in brain tumors where it is proving to be very clinically useful. The current literature and the current manuscript suggest the classifier works well for well-classifiable tumors, but it is challenging in new tumor types that are not well characterized. Rare and not well characterized sarcomas are a clinical challenge and approaches aimed at improving our understanding of these tumors and potential therapeutic approaches is much needed and valuable. The current manuscript uses a random forest machine learning classification algorithm similar to S. Peter Wu et al in "DNA Methylation based classifier for accurate molecular diagnosis of bone sarcomas." JCO Precision Oncology 2017. It seems the advantage of this approach is that it handles high dimensional data well, fast to train and prediction speed is fast. The biggest challenges with this approach can be with the tendency to overfit and therefore the number of samples they are analyzing is a strength to test the algorithm as previously much smaller datasets were used. It seems to be one of the largest datasets classified to date. I think this is a big strength of the current manuscript. However, it is not clear how methylation array adds compared to approaches focused on novel fusion discovery or other epigenetic investigations beyond methylation. The methods used are detailed but would benefit by independent computational review. An additional potential drawback of the algorithm is the lack of interpretability to the model. It is useful to be able to classify these highly diverse but not well characterized subset of sarcomas. However, based on this classification it seems further information would be useful in order to develop a deeper understanding of these tumor types, their potential cell of origin and any clinically relevant information. We are not yet using this approach clinically on all patients but in times of diagnostic uncertainty it would seem to be ideally implemented but in those cases in the less well characterized categories the ability to thoughtfully parse these rarest of different tumor types and feel confident in assigning a new diagnosis and treating accordingly is not clear. How do the authors propose to make this classifier more useful in the clinically relevant rare sarcomas is not clear.

It seems if this challenge could be improved, it could be a useful addition to other approaches to classify sarcomas including fusion search and identification of novel fusions.

In terms of the clinical annotation when there was a change in diagnosis little detail is provided on what other measures were used to validate the change in diagnosis and when a diagnosis was not changed as mentioned in the text lines 296-298 why was this the case. There were more cases 26/29 that although the classifier reclassified the diagnosis remained "open" and this is a current limitation as the algorithm provides classification potentially, but it seems not further interpretable data that would help add utility in managing and understanding these tumors. If this is not the case, more detail

on how this classification would make managing and understanding these tumors better would be great to be included and if the methylation data is more useful or more readily available than other epigenetic analyses.

Greater insights into sarcomas provided by this classifier would be good to highlight. Despite the seemingly herculean effort to perform this analysis on this broad a scale it is still not clear how estimated diagnosis gets you closer to identifying and characterizing specific rare sarcoma subtypes and most importantly managing these tumors. However, in brain tumors the classifier data seems to be further developed and may be clinically more useful so perhaps this large dataset will contribute to the development of this as a clinically useful tool. It would be beneficial at least at this stage to better understand how this tool could advance the understanding of these tumors and would new therapeutic approaches be generated based on these analyses. More details into this aspect would be very helpful especially in the subsets that are reclassified and are not well characterized.

Reviewer #1 (Remarks to the Author): Expert in sarcoma

The study by Koelsche et al presents a DNA methylation-based classifier of soft tissues and bone tumors. The 'sarcoma classifier' was trained on methylation profiles from 1,077 tumor samples from a reference sarcoma cohort. Methylation profiling was done using the EPIC and 450k Illumina arrays. Unsupervised hierarchical clustering and t-SNE analysis revealed 62 tumor-methylation classes belonging to 54 histological types. Using this dataset and a Random Forest machine learning classification, the authors developed the sarcoma classifier similarly to what they have done previously for tumors of the central nervous system (Capper et al., 2018). The classifier was validated on a set of 428 additional tumor samples coming mostly from relapsed/refractory tumors. Finally the authors used the DNA methylation profiles for detecting copy number alterations in the corresponding tumor samples.

Given that the authors have published a number of papers with similar focus previously (Capper et al., Nature, 2018; Koelsche et al., Clin Sarcoma Res, 2019; Koelsche et al., Journal of Clin Res and Clin Onc, 2019; Koelsche et al., Modern Pathology, 2018) – with the current manuscript having an almost identical figure lay out as the Capper et al paper – the added value of the work presented herein should be further emphasised.

What is the highlight of the presented work?

- New, better classifier?
- Identification of novel sarcoma types/subtypes?
- New diagnostic tool that could complement/replace current tumor-type specific molecular testing?

We thank reviewer #1 for the critical comment on the aim of our study. The sarcoma classifier is a completely new classifier, which has no overlap with the brain tumor classifier except for the conceptual background of the classifier algorithm. With this classifier now sarcomas can be diagnostically addressed. Given the estimated rate of missed sarcoma diagnoses approximating 20% this is of great interest and potential clinical impact. The DNA-methylation based sarcoma classifier represents an entirely novel approach of diagnosing these tumors and will complement current diagnostic procedures.

The potential of the presented sarcoma classifier as a diagnostic tool may be the most interesting/relevant aspect. Also, given that currently a lot of diagnostic centers perform RNA-seq for all newly diagnosed sarcomas, it would be interesting to assess how these two approaches perform in comparison.

We fully agree with reviewer #1 that the aim of this study was to develop a tool for sarcoma diagnostics.

Specific comments:

1. Line 199: 'recognised' may not be so precise. Replace with 'classified'

We replaced it as suggested.

2. Method section corresponding to the analysis for Fig 1 is incomplete. The description of hierarchical clustering is missing. What do the authors mean by 'iteratively analysed'? How was the cluster number chosen? Why a minimum of seven cases was used (why not more or less)? How was the perplexity chosen?

Unfortunately, we did not point out clearly the section in the Methods referring to t-SNE generation for Figure 1. This has been corrected. We pointed towards the methods for t-SNA analysis and we inserted:

"Hierarchical clustering: Unsupervised hierarchical clustering was performed using the 20,000 most variably methylated CpG sites across the dataset according to median absolute deviation, Euclidean distance and Ward's linkage method."

We agree, that the phrase "iteratively" was confusing and, therefore, deleted it throughout the manuscript.

We clarified on the number of seven cases by adding:

"[...] seven cases were required for defining a methylation class, which empirically proved sufficient for training a classifier and allowed prediction^{14,15}. Unsupervised clustering, respecting the minimal number of seven cases per group, led to the designation of [...]"

3. Lines 386 – 391: This paragraph is difficult to read – not correct English. Please rephrase/restructure.

We added a preposition relevant for understanding this sentence and bracketed some of the supplementary information.

4. Figure 1b – Legend with sarcoma type abbreviations is really hard to read. Perhaps present this information as a table/figure. Or at least tumor types can be listed in alphabetical order.

We agree with reviewer #1 that this information is relevant for the paper. The information is given in Supplementary Table 2. The designations are sorted according to the WHO book (2013) on soft tissue and bone tumours.

5. Line 382: We aimed at a tumor cell content of >70%. However, according to Extended Figure 2f the predicted tumor purity for most samples is much lower.

We thank reviewer #1 for bringing up this important point. To point towards this discrepancy, we added the following sentence to the "Sample selection and quality control" paragraph in the methods section:

" However, determining tumour cell content by random forest regression demonstrated that this goal was not reached for many samples⁴¹. "

6. Line 241/242 (Extended data Figure 2) – it is not clear what it is meant that potential confounding factors were excluded? Which factors?

We specified "potential confound factors". The sentence now reads:

"[...] and potential confounding factors such as sex, patients' age, type of material, type of array and tumour purity were excluded [...]"

7. Figure 1. The authors could explain a bit better what they mean by the four WHO categories. Do they mean the types of sarcomas based on WHO classification?

We agree with reviewer #1, the four categories were not explained in the main text. The passage now explains the four categories and reads as following:

"Category 1 represents methylation classes equaling a WHO entity. Category 2 represents methylation classes corresponding to a subgroup of a WHO entity. Category 3 represents methylation classes that combine WHO entities. Category 4 represents methylation classes of novel entities which are not yet defined by the WHO classification (Figure 1a)."

8. The task of the classifier is to split samples in methylation classes that were defined by unsupervised learning on the same dataset. For every sample that is tested in the cross-validation step of the Random Forest, the algorithm predicts to which of the 65 classes it belongs. However, these 65 classes have been defined using methylation profiles of all samples, including the profile of the tested sample. It should therefore be not surprising that the performance is high (0.999), as in a way, what is tested is the capability of the RF classifier to reproduce what the unsupervised learning did using the same data. It should be discussed if and how this biases the performance metrics, and if and why this is still a reasonable comparison. This does not apply to the validation set, which is independent. Also, the methylation classes are biologically meaningful, as seen by the large overlap to WHO classification, but this issue should at least be discussed regardless.

We agree that for the establishment of some methylation classes the t-SNE analysis played an important role. For these cases the same methylation data that was used for the unsupervised analysis was also used to train the supervised classification model on. This can be regarded as 'data leakage' and may theoretically result in overly optimistic performance metrics estimated by the cross-validation. However, not all methylation classes have been solely defined by an unsupervised analysis and for most cases also other clinical parameters played an important role, which is also reflected by the strong overlap with WHO classes. For these classes the t-SNE just confirmed that they are molecular distinct to other entities. That is why we think that the potential 'data leakage' bias is relatively small and the performance metrics estimates resulting from the cross-validation are still valid estimates. Moreover, as stated by the reviewer, the performance has also been assessed in an independent test data set.

For clarification we expanded to:

"Cross-validation, an internal performance metric¹⁵, of the sarcoma classifier [...]"

9. Extended Figure 1a: λ Youden is shown as a line (not red dot)

This was adjusted accordingly (in Extended Data Figure 3a).

10. Validation cohort: Are all classes evenly included in the validation cohort (an overview statement would be informative). Also, reference set is mostly based on diagnostic samples, whereas validation cohort is mostly relapsed tumors. Is there any overlap between diagnostic and relapsed tumors (coming from the same patient)?

Done as suggested. We added to the “Sample selection” paragraph in the methods section:

“The validation set included sarcomas enrolled in the INFORM, NCT-MASTER, PPT and MNP2.0 studies²⁸⁻³⁰. Rare sarcoma entities have not been over-represented. However, availability determined inclusion resulting in overrepresentation of high-grade sarcomas in the validation set.”

Genotype checks (genotype information read out on the methylation array) were performed to rule out any patient overlap either within or between the cohorts. Each sample used in our study belongs to a different patient. We added the following sentence to the Sample selection paragraph under the Methods section at the beginning:

“All samples of the reference and validation set are from individual/different patients.”

11. Line 296: Typo mistake – 26/59? What does ‘remain open’ mean?

Thanks, we corrected this typo accordingly to 26/59.

We clarified and replaced the phrase “remained open”:

“In 26/59 cases the discrepancy between histological diagnosis and classifier prediction could not be resolved due to lack of entity specific mutations.”

12. Figure 3 is not easy to follow. The 428 samples are relapsed/refractory samples. Why the top bar says Diagnostic samples? Where the 59 samples discussed in line 294 fit best in the figure?

We remodeled Figure 3 for a more intuitive understanding. “Diagnostic samples” was replaced by “Validation cohort”, the fields where the 59 discussed samples assigned to are now indicated accordingly.

13. It would be interesting to get an idea of the classification performance also on the samples that do not reach the 90% certainty threshold. This information is perhaps provided in Extended figure 3 but it is not discussed.

We agree with this notion. Calibrated classifier scores between 0.6 and 0.9 may often indicate correct prediction. However, in order to minimize false predictions, we adopted a stringent cutoff at 0.9. Readers of the paper can derive this information from Supplementary Table 3.

14. How is the performance for cases in which the classifier is very sure? E.g. at 99% certainty? What is the distribution of certainties in cases that the classifier did and did not agree with the diagnosis?

Calibrated prediction scores do not correspond to percentages. This can be seen in Extended Data Figure 3c and 3b. 3c depicts the rare scores, which always have a certain percentage of false predictions, which also is reflected in the calibrated score.

In the validation set the mean calibrated score for concordant prediction was 0.992, for discrepant prediction 0.998 and for misleading prediction (n=4) 0.964. We do not consider this information relevant and, therefore, did not amend the manuscript.

15. Given that CNAs have diagnostic relevance (as indicated in line 311) why CNA data are not included in the classifier? It may be interesting to compare its performance with vs without CNAs.

Reviewer #1 raises an important point. The classifier is based on tissue differentiation reflected by DNA methylation. The different data structure of CNV is not compatible with our algorithms. Furthermore, independence from CNV probably is important. A methylation signature, which is composed of thousands of CpG values, is more specific. CN hallmark alterations for pathognomic for, but not specific to certain entities, e.g. a chromosomal amplification region 12q13-15 covering MDM2, might otherwise strongly bias to diagnoses with such alteration, in this example well/dedifferentiated liposarcoma or intimal sarcomas, among others. Therefore, the DNA methylation-based classifiers (CNS tumours and Sarcomas) are independent from copy number alterations. However, as mentioned in the manuscript, we consider copy number profiles as valuable information, that's why we provide the CN profile in the molecular report (Extended Data Fig. 6). The most common copy number alterations encountered in the methylation classes are pointed out in the methylation class summary (Supplementary Table 2).

16. Lines 348-349: 'portion of the unrecognized cases' - it is not clear which cases the authors mean.

We specified:

"This does account for a portion of the 106/428 unrecognized cases exhibiting a calibrated score < 0.9."

17. Lines 339-340: The illumina methylation arrays do not only cover promoter regions. The authors comment that their study focuses on CpG sites localized on promoter regions. Have they tested which of the CpG sites are the most informative for their classifier? They could add an additional analysis to test this. They could group the CpGs based on those found on core promoters, extended promoters, DNase I sites, gene bodies, regulatory elements etc and run the classifier using only a set of the CpGs and compare it to the performance of the classifier when they use all CpGs. This analysis may contribute to biological understanding for different sarcoma types.

As described in reference 14 (Capper et al., doi:10.1038/nature26000) the 10,000 most informative CpGs were selected by applying first a RF for feature selection to calculate the RF permutation variable importance measure. Then, the second final RF was trained using only the 10,000 CpGs with highest variable importance. The permutation variable importance measure of the RF is a quite popular metric used to screen high-dimensional data sets, which is why the RF is also often applied in GWAS studies to screen for disease associated SNPs. We believe that this data driven feature selection will probably lead to the best possible prediction performance of our classifier, which in our opinion is more important than biological interpretability.

We removed the sentence:

"However, in our study the focus is on CpG sites localizing to the promoter regions."

We also removed 'promoter' in

"Employing DNA-methylation based categorization offers highly attractive features."

18. How does the classification performance change with tumor impurity? This is important to assess and seems relevant for the end user. e.g. split the validation data in a few bins, based on the impurity, and assess performance there. If this cannot be done, at least significance should be assessed in Fig 5b. Also, there seem to be correlations between tumor type and impurity (Extended Figure 2); does this affect the classifier?

We thank the reviewer for bringing up this important point. Purity is relevant for DNA-methylation based classification and its effect might differ on tumour subtypes. Unfortunately, the case numbers of the different subtypes in the validation cohort are too low for testing this hypothesis. However, we plotted tumour purity against the classifier performance for conventional osteosarcomas, which we suspected as purity dependent subtype, and don't see any correlation. This plot is now shown in Figure 5c. We also added means to Figure 5b and modified the discussion to emphasize that future studies are required to elucidate the effect of tumour purity on classifier performance. We added under results:

“[...] were contaminated with a higher amount of non-neoplastic cells than estimated by histological examination, although the mean value for tumour cell purity of 47,4% in non-classifiable cases was only slightly lower compared to 51,3% in classifiable cases (Figure 5).”

We restructured the according discussion part, which now reads:

“This circumstance might have contributed to classifier output scores lower than the cut off score of 0.9, consequently prompting the tumour evaluation as unclassifiable. The effect of tumour cell purity on the classifier performance is likely to be dependent on the sarcoma subtype (Figure 5). Future studies with larger case numbers are required to elucidate the effect of tumour purity on classifier performance.”

19. Discussion: What is the advantage of methylation based analysis (in addition to or instead of pathological assessment) in the end? Why should the end user consider doing methylation assays on top? It could uncover misclassifications, potentially altering treatment? However, at the moment it seems unsure if we could trust methylation enough to re-classify the tumor without additional genetic evidence for the new classification. So why not just do genetic classification on top of pathology, and save the time/money of methylation? How about RNA-seq that is currently used in clinics (at least for fusion driven sarcomas)? What are next steps to establish this in the clinic?

We agree with reviewer #1 that molecular tumour classification will significantly contribute to a more precise tumour definition, which is necessary to reduce the relatively high misclassification rate of sarcomas. The advantage of DNA methylation profiling is its independence from genetics, meaning that a sarcoma with a specific epigenetic signature does not necessarily carry a specific genetic alteration.

We are convinced of the tremendous power of methylation-based classification. In the brain tumor field methylation-based classification is introduced in WHO grading and is essential for some and desirable for many entities. We expect similar impact on the sarcoma field.

20. Classifier score calibration: REF 15 says linear SVMs in combination with MR worked best. Why not use that here?

We thank reviewer #1 for bringing up this is an interesting point. In our opinion a similar benchmark study like the one presented in reference 15 (Maros et al., 10.1038/s41596-019-0251-6) should be performed when planning to train future versions of the presented sarcoma classifier. However, we already started to work on the classifier presented here using the RF (reference 14 (Capper et al., doi:10.1038/nature26000)) workflow before the reference 15 was finalized. Also, the differences in performances between the Top 10 ML workflows presented in reference 15 are quite low, they all perform well and are in our opinion more or less exchangeable. Moreover, if they would be benchmarked on a different data set, like this one, the ranking of the workflows will likely change.

21. Lines 446-448: does this mean that only samples which reach a threshold of 0.9 are taken into account for calculating the Youden index? Where do the 500 samples come from? Training or test set?

We agree that this description is unclear. We restructured the complete paragraph separating the cross-validation from the parameter tuning of the calibration model and added Extended Data Figure 6 which shows the CV scheme in more detail (also see comment 22). For each of the 500 iteration of this resampling approach 30% of the training data was used to calculate the Youden index at a threshold of 0.9. The 500 samples are 500 random samples (data sets) of raw scores sampled without replacement. The calibration models are always fitted on training data raw scores to calibrate test data raw scores.

22. Classifier score calibration (page 18): a scheme should be provided, and the text structured in a way that makes it is easier to follow. The scheme should include the complete CV scheme, and a visualization of the resampling and definition of the binary classes "classifiable" and "unclassifiable".

We added Extended Data Figure 7, which shows the CV scheme in detail.

Extended Data Figure 6

We also rephrased the “Classifier score calibration” paragraph in the methods section, which is now provided in two paragraphs (“Classifier calibration” and “Calibration model parameter tuning”).

23. Figure 4. Color code should match Figure 3. Also, lines are hard to follow.

We checked that the colour codes of Figure 3 and Figure 4 match. For better visibility we enhanced some of the connecting lines in Figure 4.

Figure 4

24. Figure 5. Most of the unclassified cases are located in the center of the plot. Is there an explanation for this?

In our experience this is typical for a t-SNE and indicates that there are no other similar samples in the data set, leaving these samples in the middle of the plot, which is the starting position in the first iteration of the t-SNE. For methylation profiles this also often indicates a bad QC, due to low DNA content, bad bisulfite conversion, contamination, low tumor purity or other reasons.

25. Extended Figure 1b: it needs better labeling - x and y coordinates / iterations are not shown.

This figure refers to t-SNE stability and shows pairwise correlation. X and y axes refer to the methylation classes. Accordingly, correlation correlates to colour code. We remodeled this Figure in order to clarify.

Reviewer #2 (Remarks to the Author): Expert in machine learning and methylation

Summary

The authors present a classification model that classifies soft tissue and bone sarcomas from 65 methylation groups. They used tSNE to identify classes that share methylation patterns. A random forests model was created to classify the 65 methylation groups using methylation data. The raw score obtained from the random forest model was then calibrated using a logistic regression model with L2 norm to obtain scores that were comparable between classes. Some of the methylation groups did not classify well, but seemed to be related to other methylation groups that also did not classify well. Similar methylation groups that did not classify well were combined into methylation class families and the the classifier was able to correctly classify cases belonging to these families to the family, if not to the specific methylation group. The classification model was developed using the same methodology as in the paper “DNA methylation-based classification of central nervous system tumours” (Capper et al. 2018).

The authors verified the model using cross-validation and were able to achieve an error rate of less than 1% once all of their adjustments were made. They also verified their model using a

validation data set. The error rate for this validation set was far greater. Twenty-five percent of the validation samples were not able to be classified by the model using the classification criterion. Of those that were classified, 61% were classified correctly. The authors investigated the cases that were not assigned a class and those that were classified incorrectly. Those that were not classified correctly were examined for subtype alterations and the additional information allowed them to correctly classify for about half of the misclassified cases. The cases that were not given a classification by the model did not cluster tightly within a methylation class in the tSNE analysis.

The authors also examined copy number variations (CNVs) and provided a table with the CNVs alterations for each methylation group, but that table was not included in the copy that I reviewed so I cannot comment on how helpful the CNV is for the overall tumor classification process.

Main concerns

The details of the classifier should be tightened up. It seems that the authors used the same method as the paper, “DNA methylation-based classification of central nervous system tumours” (I refer to this as the CNS paper below), but some of the important details are not clarified in your paper. The paper would improve if the following are clarified or if it is clarified which methodology on the CNS paper was followed exactly:

Where the number of probes that were used for the classifier trimmed? The CNS paper used variable importance to trim the number of probes down to 10,000.

Yes. As described in reference 14 (Capper et al., doi:10.1038/nature26000) the 10,000 most informative CpGs were selected by applying first a RF for feature selection to calculate the RF permutation variable importance measure. Then the second final RF was trained using only the 10,000 CpGs with highest variable importance. The permutation variable importance measure of the RF is a quite popular metric used to screen high-dimensional data sets, which is why the RF is also often applied in GWAS studies to screen for disease associated SNPs.

The following sentence was added to the “Classifier development” section:

“We used the 10,000 CpGs with highest variable importance. In addition, to address unequal class size we performed downsampling as previously described¹⁴.”

Was down-sampling used to address unequal class sizes in the reference data set? It is mentioned that down-sampling was used, but did not indicate that it was used to address unequal class size as it was in the CNS paper.

Yes, down-sampling was performed like in Capper et al. 2018. We added:

“In addition, to address unequal class size we performed downsampling as previously described¹⁴.”

What measure was used for the random forest score? The CNS paper defines the score as the proportion of times that a sample is classified as a certain methylation group in the 10,000 trees. The score can also refer to the F1 score, which is an entirely different metric. It is important that the reader is clear on how the raw score was calculated for your model.

In both manuscripts it is the proportion of times a forest votes for a class, the default score output of a Random Forest classification model, i.e. randomForest function when the argument 'type' is set to 'prob'.

More detail is needed in the description of the random forest scores used as the explanatory variables for the calibration model (logistic regression with L2 norm). In the CNS paper, it seems that the scores for a particular case are the random forest score for each methylation class. If the same method was used, there are 65 scores for each case, most of which will be 0. (I recognize that it may be less than 65 when the methylation families are taken into account). Provide some additional discussion on why 0.9 was selected as a threshold for classification.

The threshold of 0.9 was originally established for the brain tumor classifier. Here we decided that 0.9 is an easy to communicate threshold that lays between the maximum specificity threshold at 0.958 and the maximum Youden (specificity + sensitivity -1) index at 0.836. In our opinion, as the output of both classifiers is 'well calibrated' class probability estimates, an additional threshold is kind of unnecessary. The educated clinician/user should be able to judge from the class probability score alone how much he trusts a classifier prediction. However, we realized that many users still ask for a threshold and 0.9 is now kind of accepted by the community. To guarantee that this threshold of 0.9 works comparable between different classifier versions that were trained on different data sets, we established the new calibration model parameter tuning step, which tries to find penalization parameter, so that the calibrated scores perform best (maximum Youden) at a threshold of 0.9. So instead of finding for each classifier an optimal threshold we hold the threshold fix and optimize the amount of calibration w.r.t. this threshold. We deleted the last sentence of the Classifier development paragraph (“We propose 0.9 as the threshold value for the prediction of a matching class to diagnostic samples”). We inserted a new paragraph “Calibration model parameter tuning” explaining the approach to the cutoff score of 0.9.

The paper notes that the samples in the reference dataset comprised at least 70% tumor cells, but this was not the case in the validation dataset. A table or figure that includes the information about the purity of the samples in the validation dataset, the raw and calibrated scores, and whether or not the tumor was misclassified or not classified would be helpful and more clearly illustrate the effect the tumor purity may be having on the classifier. Figure 5b shows a graph of the purity of each of the samples, but does not provide the additional information needed to clearly determine the effect of the tumor purity on classification.

This topic was also brought up by reviewer #1, please see point 18. Of note, tumour cell purity and classifier calibrated score of each case of the validation set is indicated in Supplementary Table 3.

The discussion on the validation dataset states, “molecular data were screened for subtype specific alterations” and that the authors were able to correctly classify about half of the previously misclassified points using that information. Can this process be incorporated into the classifier? That is, if a tumor is misclassified, it would not be known in real life. How would the person making the diagnosis know that they should look for the subtype alterations and use that information to adjust the classifier assigned by the model? Using the numbers provided in the “Classifier performance validated in a clinical cohort” section, the model was

only able to correctly classify about 46% of the validation cases ($428 * 0.75 * 0.61 = 195$, $195/428 = 0.4575$). If other information can be used in conjunction with the classifier to get to a correct diagnosis, it should be more clearly stated.

We agree with reviewer #2 that the user should be noticed about subtype specific alterations, depending on the classifier result. Accordingly, Supplementary Table 2 provides the description for each methylation class. This description includes information about specific genetic alterations (if present) for validation, which summarized copy number alterations frequently encountered in the respective subtype and also includes clinical information like gender and age distribution etc.. This description is provided in the molecular report, the output file of the classifier (Extended Data Fig. 6). We have added the following sentence at the end of "Results".

"Molecular and clinical characteristics of the predicted methylation class are provided in a molecular classifier report (Extended Data Figure 6)."

Regarding the 46% correctly classifier samples: There may be a misunderstanding. We have 428 sarcomas in the validation set, 322 sarcomas have a score ≥ 0.9 (75%), 263 of 428 do match (61%)

The calibration model: there are many other possible ways to normalize your model besides an L2 norm. It is not clear that other norms, such as the L1 norm or a combination of L1 and L2, would yield a higher classification rate, but it may be worth investigating other normalization methods to see if the non-classification and mis-classification rates can be decreased.

Yes, especially the R-package glmnet that we used to fit the calibration model also offers a L1 penalization as well as a combination of L1 and L2, which is known as the elastic net penalization. One problem with logistic calibration models is that the RF scores, which are the explanatory variables, often already perfectly separate classes and this leads to inflated coefficient estimates. A possible way to deal with this so called 'complete or quasi-complete separation problem' is to introduce penalization terms, and in fact this is the original reason some of the penalized regression models have been developed (for example Firth's logistic regression). Here we just wanted to make use of this property of penalization terms that is to stabilize estimation in situation in which classes are perfectly separable. In addition, we think that the calibrated scores should be a function of the raw scores of all considered methylation classes and this is why we do not use the L1 penalization or elastic net here, as unlike the L2 they would perform a variable selection. But we agree with reviewer #2, in principle this could be further investigated and might also lead to an improved prediction performance.

Please expand on how the copy number profiling data contributes to diagnosis in the context of the analysis done in this paper. Did it shed some light on some of the cases that were not classified or were not classified correctly? Illustrate a case or pattern that shows the merit of this additional information. I was not able to see Extended Data Figure 4 or Supplementary Table 2 because they were not included in my review copy so such examples may be present in those resources. However, even if that information is illustrated in the table and/or figure, pointing it out with a sentence or two in your text would illustrate the value of using the CNV information in conjunction with your classifier to diagnose a tumor.

The implementation of CNV to the validation set is described in the results section “Copy number profiling of sarcomas”, where we elucidate the value of CNV in sarcoma diagnostics by 3 examples: “Frequently encountered alterations include MDM2 amplification for well-/dedifferentiated liposarcomas, MYC amplification for radiation induced angiosarcoma or segmental chromosomal deletions on chromosome 22q encompassing SMARCB1 for rhabdoid tumours“ Extended Data Figure 4 (now 5) provides information on entity specific CNVs. Supplementary Table 2 contains the information used in the validation of individual tumors. In our validation series we do not have misclassified cases with CNV providing a solution.

Figure 2: This figure shows that the classifier is classifying very well, but because it is performing well, the figure is not providing very helpful information. The misclassified areas are difficult to identify because the color is so light. The numbers would be more helpful than the heat map in this situation. You may want to summarize this information in a table. A table can show each methylation group and the misclassification rates within each group for each classification stage (raw scores and calibrated scores) and indicate how closely related the misclassified point is to the predicted methylation class. The heat map may be very informative for the validation data. This is where you have many non-classification and misclassification errors. This type of heatmap may shed some light on the nature of these errors in that context. A table would also be helpful for the validation data. Figure 3 does provide additional information in regard to what is misclassified and a table would provide more detailed information.

As suggested, we summarized the numbers belonging to Figure 2 into an additional table (Supplementary Table 4). We created a heatmap for the validation data (new Extended Data Fig 4). Numbers of previous Extended Data Fig. Fig. 4-6 shift by one position, accordingly. Detailed information on all samples in the validation cohort including initial diagnoses and classifier predictions is shown in Supplementary Table 3.

Figure 4: This graph contains a lot of good information, but is hard to digest. If possible, it would be nice if the labels on the right lined up with the labels on the left. This allows the reader to quickly observe overall patterns, in particular when cases are correctly classified or erroneously classified to a methylation class that is closely related to the true methylation class. It would also be helpful to group the labels together in the same way they are in Figure 1 rather than in alphabetical order. That is, have tumor types from the same WHO category

grouped together and incorporate the same color structure used in Figures 1 and 2 to further illustrate methylation classes that are closely related to each other.

We thank reviewer #2 for his suggestion on designing Figure 4. However, we already implemented changes on Figure 4 based on suggestions of reviewer #1. We hope the reviewers are satisfied with the updated appearance.

Figure 5: This figure is an important illustration of the data, but it is a little hard to interpret all of the relationships it is trying to show. The dark color scheme with the red and gray dots was hard to visually interpret. I also would like to see the points that were misclassified in the validation dataset on the tSNE plot. The graphic may be more clear if you have four plots that show the same tSNE clusters for the reference dataset that are grayed out, and only add one color to each plot to highlight a single feature. The first plot can show the points that were misclassified in the reference dataset with one color, the second plot can highlight the points that were misclassified in the validation dataset in another color, the third plot can highlight the points that failed to be classified in the validation set with a third color, and the fourth plot can be the points that were classified correctly in the validation set with a fourth color. If these plots are gridded together, the relationships should be easier to identify.

We modified the color scheme to make the figure more intuitive for the reader. The misclassified cases are now indicated in red. We do not wish to expand to four plots.

Minore comments

Line 221: Add a comma after “On this basis”
Line 225: Add a comma after “With this study”
Line 318: Consider a comma after profiles

We added it as suggested.

Line 296: I believe it should be 26/59 cases

We have corrected this typo (see Reviewer #1).

Reviewer #3 (Remarks to the Author): Expert in sarcoma

Reviewer #3 (Remarks to the Author): Expert in sarcoma

This paper provides a more detailed DNA methylation profiling to classify sarcomas. Unlike previous work published by this group (Koelsche C, Clinical Cancer Research 2019 and Koelsche C, Mod Pathology 2018) this classifier was created by a much larger group of diverse sarcoma tumor types. They also had a smaller but still very large set for validation. The field of DNA methylation classification is very new in sarcomas compared to the more developed data in brain tumors where it is proving to be very clinically useful. The current literature and the current manuscript suggest the classifier works well for well-classifiable tumors, but it is challenging in new tumor types that are not well characterized. Rare and not well characterized sarcomas are a clinical challenge and approaches aimed at improving our understanding of these tumors and potential therapeutic approaches is much needed and

valuable. The current manuscript uses a random forest machine learning classification algorithm similar to S. Peter Wu et al in “DNA Methylation based classifier for accurate molecular diagnosis of bone sarcomas.” JCO Precision Oncology 2017.

It seems the advantage of this approach is that it handles high dimensional data well, fast to train and prediction speed is fast. The biggest challenges with this approach can be with the tendency to overfit and therefore the number of samples they are analyzing is a strength to test the algorithm as previously much smaller datasets were used. It seems to be one of the largest datasets classified to date. I think this is a big strength of the current manuscript. However, it is not clear how methylation array adds compared to approaches focused on novel fusion discovery or other epigenetic investigations beyond methylation. The methods used are detailed but would benefit by independent computational review.

We thank reviewer #3 for his very positive feedback. While the identification of fusions in sarcomas may be highly diagnostic, it is clear that currently only half of the sarcoma entities harbor recurrent fusions. In contrast, the methylation profile is very specific for all the sarcoma entities tested in our study.

The suggestion of an independent computational review has been taken up by the editors: reviewer 2 is an Expert in machine learning and methylation.

An additional potential drawback of the algorithm is the lack of interpretability to the model. It is useful to be able to classify these highly diverse but not well characterized subset of sarcomas. However, based on this classification it seems further information would be useful in order to develop a deeper understanding of these tumor types, their potential cell of origin and any clinically relevant information. We are not yet using this approach clinically on all patients but in times of diagnostic uncertainty it would seem to be ideally implemented but in those cases in the less well characterized categories the ability to thoughtfully parse these rarest of different tumor types and feel confident in assigning a new diagnosis and treating accordingly is not clear. **How do the authors propose to make this classifier more useful in the clinically relevant rare sarcomas is not clear.** It seems if this challenge could be improved, it could be a useful addition to other approaches to classify sarcomas including fusion search and identification of novel fusions.

We agree with the reviewer the addition information may strengthen the diagnostic power of our approach. This indeed is a task for many of the next years to come (In fact, a multicenter study to integrate mutations and epigenetic data has just been proposed in frame of the Horizon 2020 Call: H2020-SC1-BHC-2018-2020; Applicants in the sarcoma section of the call include sarcoma pathologists with a very strong focus on fusion based tumor diagnosis and also our team).

In terms of the clinical annotation when there was a change in diagnosis little detail is provided on what other measures were used to validate the change in diagnosis and when a diagnosis was not changed as mentioned in the text lines 296-298 why was this the case. There were more cases 26/29 that although the classifier reclassified the diagnosis remained “open” and this is a current limitation as the algorithm provides classification potentially, but it seems not further interpretable data that would help add utility in managing and understanding these tumors. If this is not the case, more detail on how this classification would make managing and understanding these tumors better would be great to be included

and if the methylation data is more useful or more readily available than other epigenetic analyses.

We provided an encompassing set of data to the reasoning for reclassification in the supplementary table 3. We agree that not all of the cases with a discrepancy between the classical approach and methylation could be verified. There are two reasons for this: 1) lack of additional material for in depth morphological and immunohistochemical analyses and 2) lack of specific molecular alterations these tumors could have been tested for (only about half of the sarcoma entities carry specific molecular actions). We think that we have discussed this sufficiently in the manuscript.

Greater insights into sarcomas provided by this classifier would be good to highlight. Despite the seemingly herculean effort to perform this analysis on this broad a scale it is still not clear how estimated diagnosis gets you closer to identifying and characterizing specific rare sarcoma subtypes and most importantly managing these tumors. However, in brain tumors the classifier data seems to be further developed and may be clinically more useful so perhaps this large dataset will contribute to the development of this as a clinically useful tool. It would be beneficial at least at this stage to better understand how this tool could advance the understanding of these tumors and would new therapeutic approaches be generated based on these analyses. More details into this aspect would be very helpful especially in the subsets that are reclassified and are not well characterized.

Aim of this work is not as much as providing novel insights into sarcomas rather than presenting a tool which might be useful for the diagnostician. We expect that further expansion along this way will help identifying additional rare entities or subsets of sarcomas which evade current diagnostic approaches.

Reviewers' Comments:

Reviewer #1:

Remarks to the Author:

The authors have not made a substantial effort addressing all concerns raised by the reviewers. Most importantly the way they respond to several comments is not ideal.

Few examples of specific comments that the authors could have put a bit more effort addressing them:

Reviewer 1#, comment 17: The authors instead of performing the simple analysis suggested by the reviewer, they rather removed the corresponding sentence from the discussion.

Reviewer 2#, comment on Figure 4. The authors preferred to modify the figure based on a comment by reviewer 1#, hoping that reviewer 2# will be satisfied.

Reviewer 2#, comment on Figure 5. The authors did not wish to expand the figure as suggested by the reviewer.

Reviewer 3#, last comment. The authors did not address the comment because they prefer to emphasize on their classifier as a diagnostic tool and hence did not go into potential novel biological insights that could be inferred by further analyzing their data.

However, at the same time they did not make an effort adding few lines in the discussion about their classifier as a diagnostic tool in comparison to other currently used approaches such as RNA-seq (although this was a comment by all three reviewers).

Reviewer #2:

Remarks to the Author:

The author has addressed concerns.

REVIEWER COMMENTS

We thank the reviewers for their constructive comments on the manuscript. We once again carefully revised the manuscript according to their suggestions. These changes were marked in blue, former changes were kept in red.

Reviewer #1 (Remarks to the Author): Expert in sarcoma

#1 Please group the labels together in the same way they are in Figure 1 rather than in alphabetical order to Figure 4.

We changed the order as suggested.

#2 Please add four plots that show the same tSNE clusters for the reference dataset that are grayed out, and only add one colour to each plot to highlight the single feature fore Figure 5.

We changed Figure 5 as suggested.

#3 Please clarify if the CpG sites are the most informative for your classifier and perform additional analysis if necessary.

Yes, the classifier is trained on the most informative CpGs across the reference dataset. In this context, most informative means most discriminating/variable CpGs. This information was already added to the manuscript with the previous revision (under methods). We performed additional analysis to specify the distribution of these CpGs within the gene region and their regulatory feature group provided by the manifest file of the Illumina array. This information is now shown in Extended Data Figure 7.

“We used the 10,000 CpGs with highest variable importance. In addition, to address unequal class size we performed downsampling as previously described¹⁴. The distribution of these CpGs position within the gene region and their regulatory feature group are indicated (Extended Data Figure 7).”

Please discuss about the use of your classifier for rare sarcomas, as this was a common request across all reviewers.

We specified in the discussion:

“While the current version of the sarcoma classifier already includes some very rare entities, we acknowledge not to cover the entire spectrum. Analysis of additional sarcoma samples

including uploaded data, subject to permission, will further improve this tool by refining established and adding novel methylation classes. "